# The effect of type 2 diabetes genetic predisposition on non-cardiovascular comorbidities

Ana Luiza Arruda[1,2,30] ✉, Ozvan Bocher [1,3,30], Henry J. Taylor [4,5,6,30], Davis Cammann[7,30], Satoshi Yoshiji [8,9,10], Xianyong Yin [11,12], Chi Zhao[13], Jingchun Chen[7], Alexis C. Wood[14], Ken Suzuki [15], Josep M. Mercader [8,16,17], Cassandra N. Spracklen [13], James B. Meigs[8,18,19], Marijana Vujkovic [20,21,22], George Davey Smith [23], Jerome I. Rotter [24], Benjamin F. Voight [20,21,25,26,31], Andrew P. Morris [27,28,31] & Eleftheria Zeggini [1,29,31] ✉

Type 2 diabetes is associated with a range of non-cardiovascular non-oncologic comorbidities. To move beyond associations and evaluate causal effects between type 2 diabetes genetic predisposition and 21 comorbidities, we apply Mendelian randomization analysis using genome-wide association studies across multiple genetic ancestries. Additionally, leveraging eight mechanistic clusters of type 2 diabetes genetic profiles, each representing distinct biological pathways, we investigate causal links between cluster-stratified type 2 diabetes genetic predisposition and comorbidity risk. We identify causal effects of type 2 diabetes genetic predisposition driven by distinct genetic clusters. For example, the risk-increasing effects of type 2 diabetes genetic predisposition on cataracts and erectile dysfunction are primarily attributed to adiposity and glucose regulation mechanisms, respectively. We observe opposing effect directions across different genetic ancestries for depression, asthma and chronic obstructive pulmonary disease. Our findings leverage the heterogeneity underpinning type 2 diabetes genetic predisposition to prioritize biological mechanisms underlying causal relationships with comorbidities.

Worldwide, over 500 million individuals have type 2 diabetes (T2D), a number expected to exceed 1.27 billion by 2050[1]. In 2022, diabetes-related healthcare expenditures were estimated to be over $412 billion in the United States of America alone[2]. The cumulative economic strain on healthcare resources is much greater when considering the comorbidities associated with T2D. For instance, a study looking at multimorbid T2D patients found that 73% also suffered from hypertension, followed by 69% with a diagnosis of back pain, 67% of depression, 45% of asthma, and 36% of osteoarthritis[3]. While advances in treatment, such as improved blood sugar monitoring[4] and glucagon-like peptide-1 (GLP-1) receptor agonists[5], can mitigate some of the

challenges associated with T2D, its rising prevalence and diverse comorbidities call for a paradigm shift toward prevention strategies. Despite their prevalence and significant impact on quality of life[6], non-cardiovascular T2D comorbidities remain understudied compared to cardiovascular conditions. Building on the understanding of shared genetic etiology, we can identify common biological mechanisms to develop targeted interventions tailored to an individual's genetic predisposition for multimorbidity profiles[7,8].

Genetic and environmental factors are involved in the pathophysiology of T2D, a biologically heterogeneous disease characterized by the interplay of different cell-type specific mechanisms[9]. Efforts to

identify different T2D genetic profiles have yielded promising results[10,11]. Recently, the T2D Global Genomics Initiative (T2DGGI) identified 1289 genetic risk variants in a multi-ancestry genome-wide association study (GWAS) meta-analysis[11] and grouped these variants into eight non-overlapping mechanistic clusters representing distinct biological pathways. Clustering was performed based on the association profile of the T2DGGI genetic risk variants with cardiometabolic traits, yielding the following clusters: body fat ($n = 273$ variants), obesity ($n = 233$), beta cell associated with increased proinsulin (+PI) levels ($n = 91$), beta cell associated with decreased proinsulin (-PI) levels ($n = 89$), metabolic syndrome ($n = 166$), lipodystrophy ($n = 45$), liver/lipid metabolism ($n = 3$) and residual glycaemic ($n = 389$). Associations between the T2DGGI partitioned genetic risk scores (GRS) and cardiovascular outcomes, including coronary artery disease and diabetic retinopathy, have been previously described[10,11].

To move beyond associations, we evaluate causal relationships between T2D genetic predisposition and 21 non-cardiovascular non-oncologic comorbidities by employing bi-directional Mendelian randomization (MR). MR is a causal inference method that uses genetic variants as instrumental variables (IVs) to estimate the causal effect of an exposure on an outcome, leveraging the random allocation of alleles at conception to reduce confounding and reverse causation. For MR estimates to be valid, three core assumptions must hold: (1) the IVs are robustly associated with the exposure of interest; (2) the IVs are not associated with any confounders of the exposure-outcome relationship; and (3) the IVs influence the outcome solely through the exposure, not via alternative pathways (no horizontal pleiotropy). We acknowledge that violations of these assumptions pose challenges to causal inference and have implemented sensitivity analyses in accordance to the STROBE guidelines to test the robustness of our findings (Methods, Supplementary Note 1, Supplementary Note 2).

Previous studies investigating the causal links between genetic predisposition for T2D and its non-cardiovascular comorbidities have not considered genetic subtypes of disease heterogeneity[12–20]. Cluster-stratified MR analysis has been previously employed to identify specific biological mechanisms driving causal relationships and potential pleiotropic pathways[21–23]. Here, in addition to performing pairwise bi-directional MR analyses, we investigated cluster-stratified effects of T2D genetic predisposition on its comorbidities by restricting the IVs to T2DGGI genetic risk variants assigned to each mechanistic cluster identified in the latest T2DGGI GWAS meta-analysis. Our biologically informed cluster-stratified MR approach aims to identify distinct T2D genetic profiles causally associated with 21 non-cardiovascular diseases, spanning five classes: musculoskeletal, respiratory, reproductive, neuropsychiatric and ophthalmic conditions. The different T2D genetic profiles highlight potential biological pathways underlying specific comorbidity pairs.

## Results
### Genetic predisposition for T2D is a potential driver of comorbidities
We investigate causal effects between T2D genetic predisposition and 21 non-cardiovascular comorbidities (Table 1) using pairwise bi-directional two-sample MR (Fig. 1). The T2D comorbidities were selected after a comprehensive literature review of observational studies and the availability of GWAS summary statistics without the inclusion of proxy cases.

We find statistical evidence (FDR-adjusted $p$-value ($q$-value) <0.05) of a causal link between overall T2D genetic predisposition and 11 comorbidities (Fig. 2, Supplementary Data 1). Our findings show that genetic predisposition for T2D has risk-increasing effects on attention-deficit/hyperactivity disorder (ADHD) (IVW odds ratio (OR) = 1.09, $p$-value = 9.17×$10^{-16}$), carpal tunnel syndrome (CTS) (OR = 1.10, $p$-value = 1.81×$10^{-30}$), cataracts (OR = 1.04, $p$-value = 1.35×$10^{-10}$), erectile dysfunction (OR = 1.09, $p$-value = 2.54×$10^{-9}$), osteoarthritis

(OR = 1.01, $p$-value = 7.10×$10^{-3}$), polycystic ovary syndrome (PCOS) (OR = 1.12, $p$-value = 7.47×$10^{-7}$), primary open-angle glaucoma (glaucoma) (OR = 1.07, $p$-value = 1.81×$10^{-9}$) and vascular dementia (OR = 1.09, $p$-value = 1.30×$10^{-3}$). For anorexia nervosa (OR = 0.96, $p$-value = 2.25×$10^{-3}$), obsessive-compulsive disorder (OCD) (OR = 0.90, $p$-value = 2.16×$10^{-4}$) and osteoporosis (OR = 0.95, $p$-value = 8.55×$10^{-5}$), we find T2D genetic predisposition to have a protective effect. When comparing our causal estimates with results of previous MR studies investigating the effect of T2D genetic predisposition on disease risk, we find mostly consistent directions of effect (Supplementary Fig. 1, Supplementary Table 1)[12,13,15–20,24–33]. For anorexia nervosa, vascular dementia and OCD, we did not find MR studies using T2D genetic predisposition as the exposure for comparison.

In the reverse direction, we find statistical evidence of a protective effect of genetic predisposition for clinically diagnosed Alzheimer's disease (OR = 0.98, $p$-value = 4.81×$10^{-3}$), bipolar disorder (OR = 0.96, $p$-value = 6.62×$10^{-4}$) and vascular dementia (OR = 0.95, $p$-value = 3.55×$10^{-39}$) on risk for T2D (Supplementary Fig. 2, Supplementary Data 2). The identified protective effects of genetic predisposition for Alzheimer's disease and vascular dementia on T2D risk are attenuated after adjustment (multivariate MR) for high-density lipoprotein (HDL) levels, body mass index (BMI) and waist-to-hip ratio (WHR) (Fig. 1, Methods, Supplementary Note 2).

### Distinct biological mechanisms contribute to the effect of T2D genetic predisposition on comorbidities
In addition to the effects of overall genetic predisposition for T2D on comorbidity risk, we investigated the effects stratified by T2D mechanistic subtypes using the T2DGGI genetic variants assigned to each T2DGGI genetic cluster as IVs (Fig. 1). All identified causal relationships between overall genetic predisposition for T2D and its comorbidities are accompanied by at least one cluster-stratified effect, suggesting shared underlying biological pathways (Fig. 3, Supplementary Fig. 3).

We also provide evidence of causal effects of certain T2DGGI genetic clusters without the presence of a causal effect of overall genetic predisposition for T2D on five further comorbidities: Alzheimer's disease, asthma, chronic back pain, chronic obstructive pulmonary disease (COPD) and rheumatoid arthritis (Fig. 3, Supplementary Data 1). To investigate whether the observed effects are specific to the respective clusters, we performed leave-one-cluster-out MR analyses using all T2DGGI genetic risk variants except those assigned to the potential causal cluster (Methods, Supplementary Note 1). In the leave-one-cluster-out MR analyses, we find no evidence of a causal effect of T2D genetic predisposition on any of the five diseases, indicating specific biological mechanisms, represented by the causal clusters, that may underlie these relationships (Supplementary Data 3).

### The impact of T2D obesity-related pathways on anorexia nervosa, asthma and back pain
We find evidence of causal effects of the obesity cluster on 11 out of the 16 comorbidities (68.75%) causally affected by any genetic predisposition for T2D. The effects of the obesity cluster are consistently the largest across all clusters (Fig. 3, Supplementary Data 1). For asthma (OR = 1.12, $p$-value = 6.40×$10^{-4}$) and chronic back pain (OR = 1.09, $p$-value = 3.69×$10^{-6}$), we find a risk-increasing effect of the obesity cluster but not of any of the other T2DGGI genetic clusters. Similarly, for anorexia nervosa, we find a protective effect of the obesity cluster (OR = 0.91, $p$-value = 9.37×$10^{-3}$) without a further effect of another T2DGGI genetic cluster.

To validate the MR findings, we performed a phenome-wide association study (PheWAS) for each T2DGGI genetic cluster using data from the *All of Us* Research Program, an ancestrally and culturally

**Table 1 | Overview of non-cardiometabolic type 2 diabetes comorbidities GWAS**

| Disease | Genetic ancestry | N | Ncases | Ncontrols | Cases definition | | Ref. |
|---|---|---|---|---|---|---|---|
| | | | | | SR | CD | |
| Alzheimer's disease | EUR | 63,936 | 21,982 | 41,954 | | x | 80 |
| Anorexia nervosa | EUR | 72,517 | 16,992 | 55,525 | x | x | 81 |
| Asthma | EUR | 1,376,071 | 121,940 | 1,254,131 | x | x | 82 |
| | EAS | 341,204 | 18,549 | 322,655 | x | x | 82 |
| | AMR | 18,173 | 4069 | 14,104 | x | x | 82 |
| | SAS | 27,091 | 4015 | 23,076 | x | x | 82 |
| | AFR | 32,658 | 5051 | 27,607 | x | x | 82 |
| ADHD | EUR | 225,534 | 38,691 | 186,843 | | x | 83 |
| Autism | EUR | 46,350 | 18,381 | 27,969 | | x | 84 |
| Back pain | EUR | 158,025 | 29,531 | 128,494 | x | x | 85 |
| Bipolar disorder | EUR | 413,466 | 41,917 | 371,549 | | x | 86 |
| CTS | EUR | 1,239,680 | 48,843 | 1,190,837 | | x | 87 |
| Cataracts | EUR | 585,243 | 67,844 | 517,399 | | x | 88 |
| COPD | EUR | 995,917 | 58,559 | 937358 | x | x | 89 |
| | AFR | 29,682 | 1978 | 27,704 | x | x | 89 |
| | AMR | 15,086 | 1503 | 13,583 | x | x | 89 |
| | EAS | 329,733 | 19,044 | 310,689 | x | x | 89 |
| Depression | EUR | 404,529 | 170,756 | 233,773 | x | x | 90 |
| | EAS | 382,936 | 21,980 | 360,956 | x | x | 91 |
| | AMR | 377,959 | 25,013 | 352,946 | x | x | 91 |
| | SAS | 31,681 | 4505 | 27,176 | x | x | 91 |
| | AFR | 198,497 | 36,818 | 161,679 | x | x | 91 |
| Epilepsy | EUR | 82,482 | 29,944 | 52,538 | | x | 92 |
| Erectile dysfunction | EUR | 223,805 | 6175 | 217,630 | x | x | 93 |
| Glaucoma | EUR | 379,422 | 29,241 | 350,181 | x | x | 94 |
| OCD | EUR | 9725 | 2688 | 7037 | | x | 95 |
| Osteoarthritis | EUR | 826,690 | 177,517 | 649,173 | x | x | 96 |
| Osteoporosis | EUR | 488,501 | 8520 | 479,981 | | x | 97 |
| | EAS | 178,726 | 9794 | 168,932 | | x | 97 |
| PCOS | EUR | 120,023 | 5209 | 114,814 | | x | 98 |
| Rheumatoid arthritis | EUR | 97,173 | 22,350 | 74,823 | | x | 99 |
| | EAS | 173,633 | 11,025 | 162608 | | x | 99 |
| Schizophrenia | EUR | 130,644 | 53,386 | 77,258 | | x | 100 |
| | EAS | 30,761 | 14,004 | 16,757 | | x | 100 |
| | AFR | 10,070 | 6152 | 3918 | | x | 100 |
| | AMR | 4324 | 1234 | 3090 | | x | 100 |
| Vascular dementia | EUR | 466,606 | 3892 | 462,714 | | x | 101 |

The overview includes genetic ancestry defined based on the continental super populations of the 1000 Genomes Project phase 3[39] (AMR = Admixed American; AFR = African; EAS = East Asian; EUR = European; SAS = South Asia), total sample size (N), number of cases (Ncases), number of controls (Ncontrols), case definition (SR = self-reported; CD = clinically diagnosed) and the corresponding GWAS publication. (ADHD = attention-deficit/hyperactivity disorder; COPD = chronic obstructive pulmonary disease; CTS = Carpal tunnel syndrome; OCD = obsessive-compulsive disorder; PCOS = polycystic ovary syndrome).

diverse cohort that was not included in the T2DGGI GWAS meta-analysis nor in any of the T2D comorbidity GWAS[34] (Methods). Concordantly, we observe an association between the T2DGGI obesity cluster genetic risk variants and asthma, chronic back pain and eating disorder (a trait strongly associated with anorexia nervosa) in the *All of Us* cohort dataset (asthma: OR = 1.04, p-value = 1.54×10⁻⁸; back pain: OR = 1.03, p-value = 4.19×10⁻⁷; eating disorder: OR = 1.14, p-value = 2.20×10⁻⁴). Our findings suggest that obesity is the main biological mechanism driving the observed causal effect of T2D genetic predisposition on the risk of these three health conditions.

BMI can act as a confounder of the identified relationships as it is a shared risk factor of T2D and the 11 comorbidities causally affected by the obesity cluster (Methods, Fig. 1, Supplementary Fig. 4, Supplementary Data 4). Hence, we adjust the obesity cluster estimates for genetic predisposition for higher BMI through multivariable MR (Methods, Supplementary Note 1, Fig. 1, Supplementary Figs. 5–28). Following adjustment, we no longer observe an effect of T2D genetic predisposition on any disease except cataracts, CTS, and osteoporosis (Supplementary Figs. 5–7, Supplementary Data 5, Supplementary Note 1).

**The impact of T2D glucose regulation mechanisms on glaucoma, COPD, rheumatoid arthritis and erectile dysfunction**
Beta cell-related clusters drive the protective effect of T2D genetic predisposition on COPD (+ PI: OR = 0.97, p-value = 8.20 × 10⁻⁴) and rheumatoid arthritis (-PI: OR = 0.92, p-value = 1.68×10⁻³), adding genetic evidence to the potential role of beta cell dysfunction in these diseases[35–38]. In addition to the beta cell clusters, variants in the

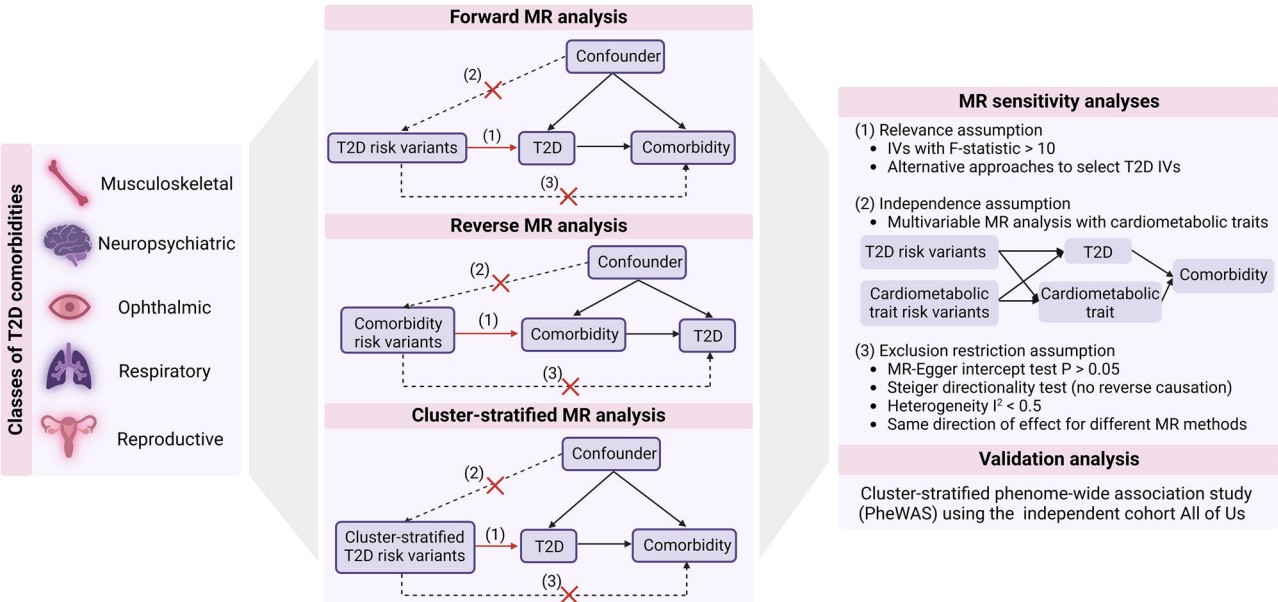

**Fig. 1 | Overview of study design.** (MR = Mendelian randomization; T2D = type 2 diabetes; IVs = genetic instrumental variables; *P* = *p*-value). Created in BioRender. Arruda, A. (2025) https://BioRender.com/rl6bj2b.

residual glycaemic cluster, which are most strongly associated with fasting glucose, also drive the risk-increasing effects of genetic predisposition for T2D on erectile dysfunction (residual glycaemic: OR = 1.12, *p*-value = 4.49×10⁻⁴; beta cell +PI: OR = 1.09, *p*-value = 4.27×10⁻³) and glaucoma (residual glycaemic: OR = 1.08, *p*-value = 7.92×10⁻³; beta cell +PI: OR = 1.11, *p*-value = 1.75×10⁻⁵). For erectile dysfunction, sensitivity analyses show statistical evidence of a robust risk-increasing causal effect of T2D genetic predisposition driven by glucose regulation clusters (Supplementary Note 1).

### The impact of T2D non-glucose-related pathways on ADHD, cataracts, PCOS, Alzheimer's disease and osteoporosis
Our cluster-stratified MR findings show that the effect of T2D genetic predisposition for cataracts is driven by the obesity (OR = 1.08, *p*-value = 1.80×10⁻⁷) and body fat clusters (OR = 1.05, *p*-value = 9.38×10⁻³). Concordantly to the risk-increasing effect of overall T2D genetic predisposition on cataracts, we observe a positive association between the T2DGGI genetic risk variants and cataracts in the *All of Us* cohort (OR = 1.08, *p*-value = 2.13×10⁻³). The protective effect of genetic predisposition for T2D on osteoporosis is driven by the metabolic syndrome (OR = 0.92, *p*-value = 3.36×10⁻⁴) and obesity clusters (OR = 0.88, *p*-value = 3.26×10⁻³). Similarly, we find negative associations in the *All of Us* cohort between T2DGGI genetic risk variants assigned to the obesity cluster and osteoporosis (obesity: OR = 0.94, *p*-value = 7.34×10⁻⁹). For PCOS, the risk-increasing effect of genetic predisposition for T2D is driven by the obesity (OR = 1.36, *p*-value = 1.90×10⁻⁷) and lipodystrophy clusters (OR = 1.29, *p*-value = 8.89×10⁻³).

For Alzheimer's disease, our findings indicate a risk-increasing effect of T2D genetic predisposition only through the body fat cluster (OR = 1.10, *p*-value = 8.28×10⁻³). Adjusting for HDL levels fully attenuates this effect (Supplementary Fig. 24, Supplementary Data 5). For ADHD, a combination of the obesity (OR = 1.24, *p*-value = 1.54×10⁻¹⁵), body fat (OR = 1.17, *p*-value = 1.15×10⁻⁸) and metabolic syndrome clusters (OR = 1.11, *p*-value = 2.42×10⁻⁴) drive the risk-increasing effect of T2D genetic predisposition. These results suggest that non-glucose-related metabolic and adiposity pathways may contribute to the link between T2D genetic predisposition and increased ADHD and Alzheimer's disease risks.

### The multifactorial impact of T2D on CTS, OCD vascular dementia and osteoarthritis
We find evidence of a causal link between all T2DGGI genetic clusters, except the beta cell clusters, and CTS (Fig. 3, Supplementary Data 1), demonstrating a multifactorial association mediated by diverse biological pathways. The strongest effects are observed for the obesity (OR = 1.23, *p*-value = 7.88×10⁻²⁶) and lipodystrophy clusters (OR = 1.20, *p*-value = 8.67×10⁻⁵). In line with these findings, T2DGGI genetic risk variants assigned to these two clusters are positively associated with CTS in the *All of Us* data (obesity: OR = 1.07, *p*-value = 7.64×10⁻¹⁰, lipodystrophy: OR = 1.04, *p*-value = 1.30×10⁻⁴) (Supplementary Data 6). The obesity (OR = 0.77, *p*-value = 8.57×10⁻⁵) and residual glycaemic clusters (OR = 0.84, *p*-value = 8.58×10⁻³) drive the protective effect of T2D genetic predisposition on OCD. The risk-increasing effect of T2D genetic predisposition on vascular dementia is driven by the obesity cluster (OR = 1.18, *p*-value = 1.13×10⁻³) and by the beta cell cluster associated with increased proinsulin levels (OR = 1.13, *p*-value = 4.71×10⁻³).

Osteoarthritis is the only disease for which we find divergent patterns of potentially causal effects across the T2DGGI genetic clusters. The identified risk-increasing effect of genetic predisposition for T2D on osteoarthritis is driven by the obesity cluster (OR = 1.15, *p*-value = 1.03×10⁻³⁷). Supporting our findings, the T2DGGI genetic risk variants assigned to the obesity cluster are positively associated with osteoarthritis risk in the *All of Us* cohort (OR = 1.06, *p*-value = 2.04×10⁻¹⁵). In contrast, genetic predisposition for T2D restricted to the beta cell clusters is associated with reduced osteoarthritis risk (+ PI: OR = 0.97, *p*-value = 1.70×10⁻⁴, -PI: OR = 0.96, *p*-value = 7.21×10⁻³). All estimates of T2D genetic predisposition on osteoarthritis risk are attenuated after adjusting for the effects of BMI, waist-to-hip ratio, and subcutaneous adipose tissue volume (Supplementary Figs. 11,21,25, Supplementary Data 5).

### Potential causal effects of genetic predisposition for T2D vary across genetic ancestry groups
We examined the MR findings within each genetic ancestry group prior to meta-analysis. (Supplementary Fig. 29). Although genetic ancestry is on a continuous scale, here we use the continental super populations defined by the 1000 Genomes Project phase 3[39], i.e. individuals

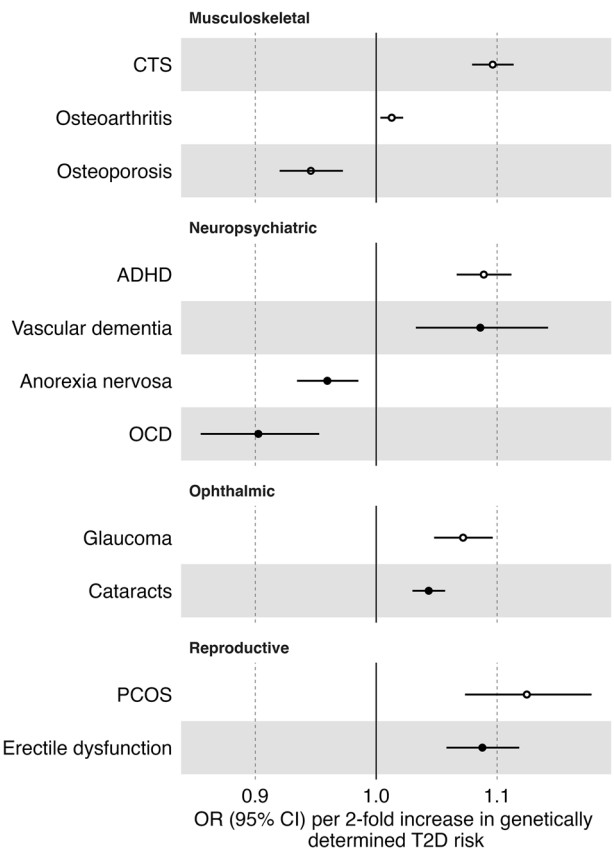

**Fig. 2 | Forward Mendelian randomization results.** Results of two-sample Mendelian randomization (MR) analysis of genetic predisposition for type 2 diabetes (T2D) on non-cardiovascular comorbidity risk for the causal relationships ($q$-value < 0.05). Causal estimates are expressed as the odds ratios (ORs) for each comorbidity per doubling (2-fold increase) in genetically determined dichotomous T2D risk. Points represent MR causal estimates derived from summary statistics (ORs, measure of center) and error bars denote 95% confidence intervals (CI). Filled circles mark estimates that passed the sensitivity analyses to assess the validity of the MR assumptions. Sample size of the T2D GWAS meta-analysis used as exposure datasets: 2,107,149 controls and 428,452 cases. Sample sizes of the GWAS used as outcome datasets: CTS ($n$ = 48,843 cases, $n$ = 1,190,837 controls), osteoarthritis ($n$ = 177,517 cases, $n$ = 649,173 controls), osteoporosis ($n$ = 18,314 cases, $n$ = 648,913 controls), ADHD ($n$ = 38,691 cases, $n$ = 186,843 controls), vascular dementia ($n$ = 3,892 cases, $n$ = 462,714 controls), anorexia nervosa ($n$ = 16,992 cases, $n$ = 55,525 controls), OCD ($n$ = 2688 cases, $n$ = 7037 controls), glaucoma ($n$ = 29,241 cases, $n$ = 350,181 controls), cataracts ($n$ = 67,844 cases, $n$ = 517,399 controls), PCOS ($n$ = 5209 cases, $n$ = 114,814 controls) and erectile dysfunction ($n$ = 6175 cases, $n$ = 217,630 controls). Abbreviations: CI confidence interval, CTS Carpal tunnel syndrome, ADHD attention-deficit/hyperactivity disorder, OCD obsessive-compulsive disorder, PCOS polycystic ovary syndrome.

genetically similar to Africans (AFR), Europeans (EUR), East Asians (EAS), South Asians (SAS), and Admixed Americans (AMR). We observe divergent patterns of association between T2D genetic predisposition and asthma, COPD and depression risk among different genetic ancestry groups (Fig. 4, Supplementary Data 7). Although no statistical evidence of a causal effect of T2D genetic predisposition on depression was found in the meta-analysis across global populations, a protective effect driven by the obesity cluster was found in EAS. On the contrary, in EUR we find evidence of a risk-increasing effect of T2D genetic predisposition on depression linked to the obesity and body fat clusters (Fig. 4, Supplementary Data 7). All T2DGGI genetic risk variants and those assigned to the obesity cluster are positively associated with depression in the *All of Us* cohort (all: OR = 1.03, $p$-value = 4.23×$10^{-5}$, obesity: OR = 1.05, $p$-value = 1.54×$10^{-5}$).

We find a protective effect of T2D genetic predisposition on COPD in AMR, contrasting a risk-increasing effect in EUR. For EUR, we find additional risk-increasing effects from the body fat and obesity clusters and a protective effect from the beta cell +PI cluster (Fig. 4, Supplementary Data 7). In addition to COPD, divergent effects across T2DGGI genetic clusters are observed in osteoarthritis. For both diseases, the risk-increasing effects of the obesity cluster oppose the protective effects of beta cell-related clusters. Results in EAS and SAS show a protective effect of T2D genetic predisposition on asthma. In EAS, the effect is driven by the beta cell and the residual glycaemic clusters. Conversely, in EUR, we identify a risk-increasing effect driven by the obesity cluster.

## Discussion

T2D is a leading global health concern that impacts individuals and healthcare systems. Effective prevention strategies for T2D and its comorbidities require a deeper understanding of the shared biology underlying these relationships. Here, we sought to identify non-cardiovascular, non-oncologic diseases causally affected by T2D genetic predisposition and investigated the biological mechanisms underlying the identified causal effects. Our results show that T2D genetic predisposition is a driver of comorbidities, rather than the risk of T2D being causally affected by genetic predisposition for its comorbidities. Moreover, T2D genetic predisposition is primarily linked to risk-increasing effects on its comorbidities, reinforcing the role of genetic burden to T2D as a risk factor. The obesity cluster drives most of the observed associations and shows the strongest effects, supporting the well-known role of obesity as a common risk factor for multiple chronic diseases[40].

Despite potential differences in phenotype definition and selection of IVs, our genetic findings align with previous MR studies investigating the causal effect of overall genetic predisposition for T2D on comorbidity risk[12,13,15–20,24–33] (Supplementary Fig. 1). Additionally, our findings align with observational studies for many of the identified associations using overall genetic predisposition for T2D, including cataracts[41], glaucoma[42,43], PCOS[44,45] and erectile dysfunction[46–48]. For instance, cohort studies have shown a decreased risk of T2D in anorexia nervosa patients[49] and a longitudinal observational study using the Polish National Health Fund data showed a decreasing trend of anorexia nervosa prevalence in T2D patients[50]. The risk of fractures, a proxy for osteoporosis, is lower in T2D patients than in healthy controls[51], in line with the identified protective effect of T2D genetic predisposition on osteoporosis.

In our study, we go beyond T2D genetic predisposition as a homogeneous phenotype by applying cluster-stratified MR and, hence, prioritize the biological mechanisms underlying the identified causal relationships. For instance, we show that glucose regulation mechanisms contribute to the associations between T2D genetic predisposition and erectile dysfunction and glaucoma, which aligns with previous genetics-based studies using glycaemic traits data[15,18,25,52]. Obesity has been positively associated with cataracts in a meta-analysis of over 1.6 million individuals[53]. Concordantly, our findings highlight obesity as one of the mechanisms underlying the causal effect of T2D genetic predisposition on cataract risk. For osteoarthritis, we find evidence of diverging cluster-stratified effects of genetic predisposition for T2D, validating the previously shown opposite association between both diseases in addition to shared risk-increasing obesity mechanisms[32].

Previous MR studies have not found evidence of a causal link between T2D genetic predisposition and osteoarthritis[31,32]. By leveraging the largest and most recent GWAS datasets for both diseases, we are now able to identify a significant risk-increasing effect of T2D genetic predisposition on osteoarthritis. This effect is attenuated after adjusting for adiposity-related traits such as BMI, WHR and subcutaneous adipose tissue volume. This indicates that these shared risk

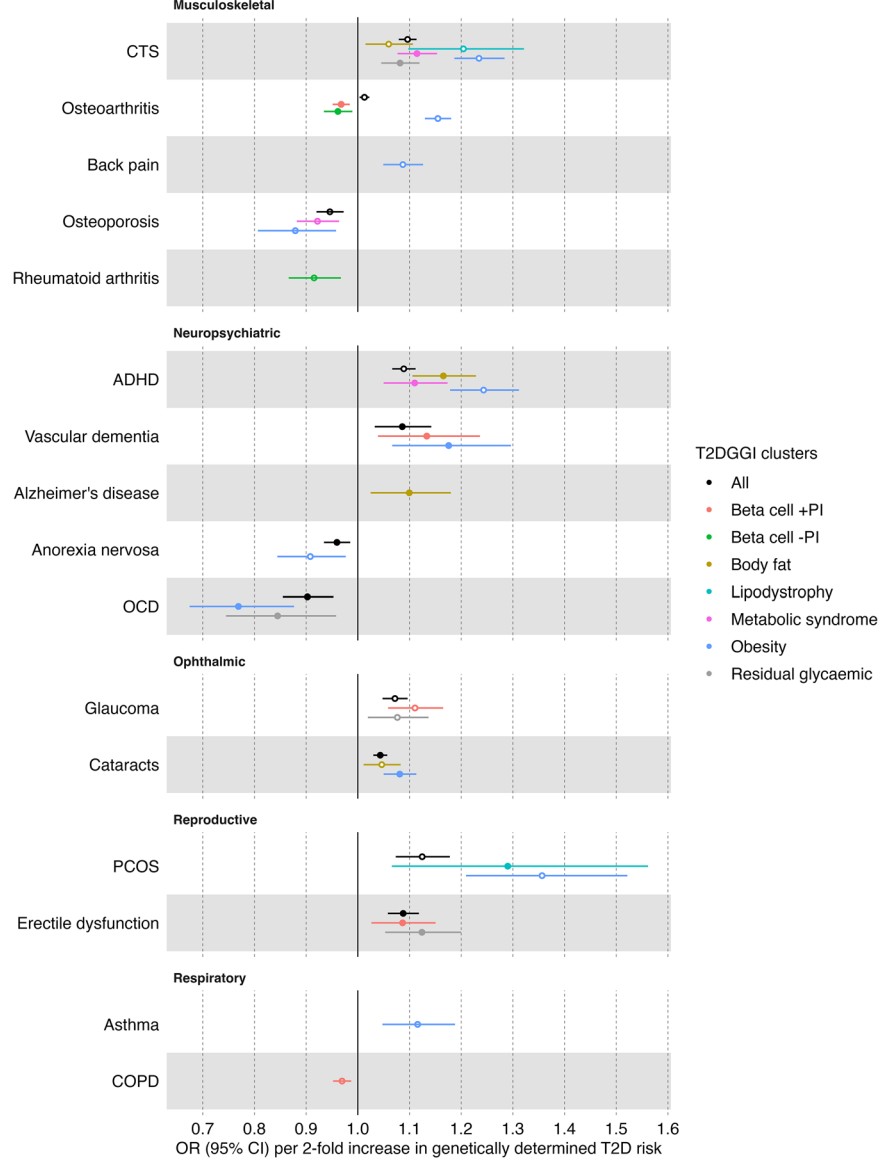

**Fig. 3 | Cluster-stratified Mendelian randomization results.** Results of cluster-stratified two-sample Mendelian randomization (MR) analysis of genetic predisposition for type 2 diabetes (T2D) on non-cardiovascular comorbidities risk for the causal relationships (q-value < 0.05). Causal estimates are expressed as the odds ratios (ORs) for each comorbidity per doubling (2-fold increase) in genetically determined dichotomous T2D risk. Points represent MR causal estimates derived from summary statistics (ORs, measure of center) and error bars denote 95% confidence intervals (CI). Filled circles mark estimates that passed all sensitivity analyses to assess the validity of the MR assumptions. Sample size of the T2D GWAS meta-analysis used as exposure datasets: 2,107,149 controls and 428,452 cases. Sample sizes of the GWAS used as outcome datasets: CTS ($n$ = 48,843 cases, $n$ = 1,190,837 controls), osteoarthritis ($n$ = 177,517 cases, $n$ = 649,173 controls), back pain ($n$ = 29,531 cases, n = 128,494 controls), osteoporosis ($n$ = 18,314 cases, $n$ = 648,913 controls), rheumatoid arthritis ($n$ = 33,375 cases, $n$ = 237,431 controls), ADHD ($n$ = 38,691 cases, $n$ = 186,843 controls), vascular dementia ($n$ = 3,892 cases, $n$ = 462,714 controls), Alzheimer's disease ($n$ = 21,982 cases, $n$ = 41,954 controls), anorexia nervosa ($n$ = 16,992 cases, $n$ = 55,525 controls), OCD ($n$ = 2688 cases, $n$ = 7037 controls), glaucoma ($n$ = 29,241 cases, $n$ = 350,181 controls), cataracts ($n$ = 67,844 cases, $n$ = 517,399 controls), PCOS ($n$ = 5209 cases, $n$ = 114,814 controls), erectile dysfunction ($n$ = 6175 cases, $n$ = 217,630 controls), asthma ($n$ = 153,624 cases, $n$ = 1,641,573 controls) and COPD ($n$ = 81,084 cases, $n$ = 1,289,334 controls). Abbreviations: T2DGGI Type 2 Diabetes Global Genomics Initiative, CI confidence interval, PI proinsulin, CTS Carpal tunnel syndrome, ADHD attention-deficit/hyperactivity disorder, OCD obsessive-compulsive disorder, PCOS polycystic ovary syndrome, COPD chronic obstructive pulmonary disease.

factors explain at least partially the effect of T2D genetic predisposition on osteoarthritis risk. We also observe contrasting cluster-specific effects: genetic variants from the T2DGGI obesity cluster have a risk-increasing effect on osteoarthritis, while variants assigned to beta cell function-related T2DGGI clusters show a protective effect on osteoarthritis risk. These findings align with prior observations of opposing associations at shared genetic loci between the two conditions[32], and highlight the value of mechanistic decomposition of polygenic risk.

We identify protective effects of genetic predisposition for Alzheimer's disease and vascular dementia on T2D risk that do not show evidence of horizontal pleiotropy. For vascular dementia, the reverse MR analysis was driven primarily by a single genome-wide significant variant at the *APOE* locus, a well-established vertical pleiotropy region with roles in both lipid metabolism and neurodegeneration[54]. The protective effect observed may thus reflect *APOE*-mediated lipid regulation. This is further supported by the attenuation of the identified effect after adjustment for genetic predisposition for higher HDL

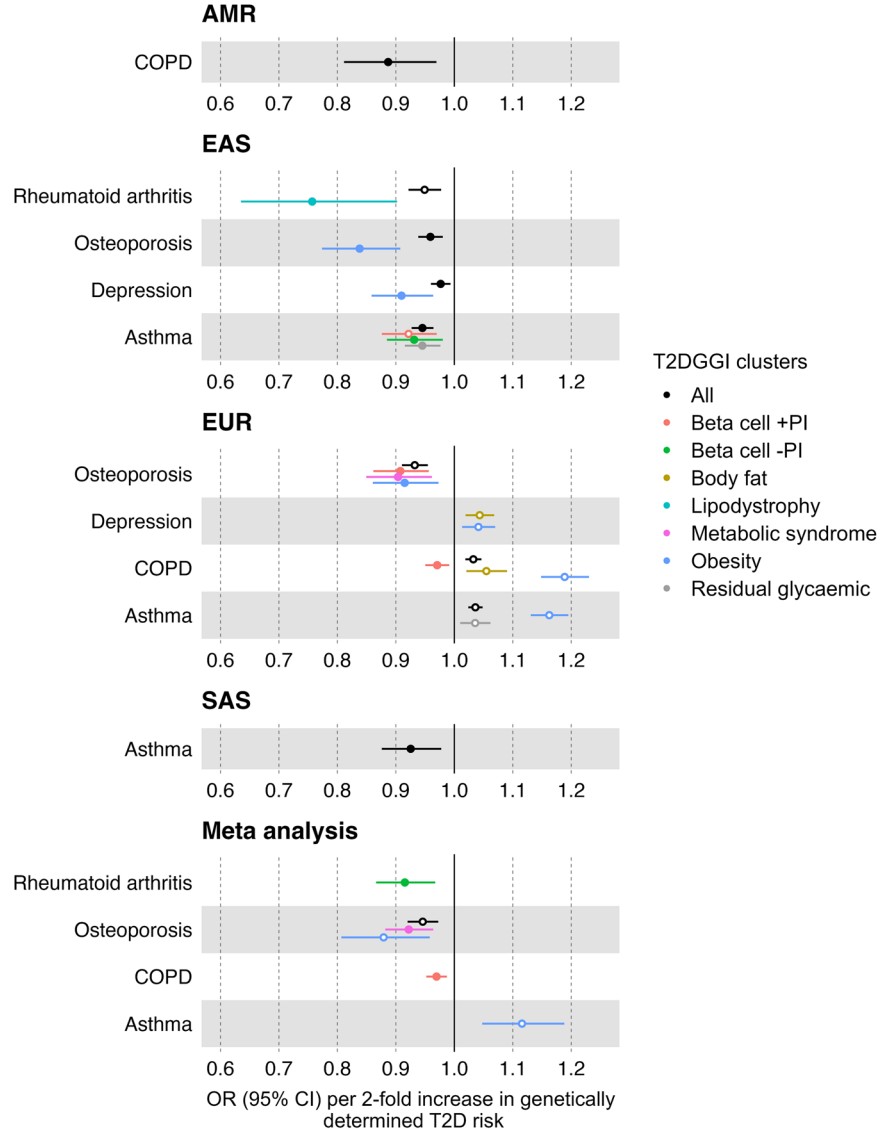

**Fig. 4 | Single-ancestry Mendelian randomization results.** Mendelian randomization (MR) results for different global genetic ancestry groups using cluster-stratified genetic predisposition for T2D and risk for T2D non-cardiovascular comorbidities (EUR=individuals genetically similar to Europeans, EAS=individuals genetically similar to East Asians) for the causal relationships (q-value < 0.05). Causal estimates are expressed as the odds ratios (ORs) of each comorbidity per doubling (2-fold increase) in genetically determined dichotomous T2D risk. Points represent MR causal estimates derived from summary statistics (ORs, measure of center), and error bars denote 95% confidence intervals (CI). Filled circles mark robust estimates that passed all sensitivity analyses. Sample size of the T2D GWAS meta-analysis used as exposure datasets: AMR = 29,375 cases, n = 59,368 controls; EAS = 88,109 cases, n = 339,395 controls; EUR = 242,283 cases, n = 1,569,734 controls; SAS = 16,832 cases, n = 33,767 controls. Sample sizes of the GWAS used as outcome datasets: COPD (AMR: n = 1503 cases, n = 13,583 controls; EUR: n = 58,559 cases, n = 937,358 controls; meta-analysis: n = 81,084 cases, n = 1,289,334 controls), osteoporosis (EAS: n = 9794 cases, n = 168,932 controls; EUR: n = 8520 cases, n = 479,981 controls; meta-analysis: n = 18,314 cases, n = 648,913 controls), rheumatoid arthritis (EAS: n = 11,025 cases, n = 162,608 controls; meta-analysis: n = 33,375 cases, n = 237,431 controls), asthma (EAS: n = 18,549 cases, n = 322,655 controls; EUR: n = 121,940 cases, n = 1,254,131 controls; SAS: n = 4015 cases, n = 23,076 controls; meta-analysis: n = 153,624 cases, n = 1,641,573 controls), and depression (EAS: n = 21,980 cases, n = 360,956 controls; EUR: n = 170,756 cases, n = 233,773 controls). Abbreviations: T2DGGI Type 2 Diabetes Global Genomics Initiative, CI confidence interval, PI proinsulin, COPD chronic obstructive pulmonary disease.

levels. The observed attenuation suggests that the effect of genetic predisposition for both dementia types on T2D risk is at least partially explained by HDL levels. In addition, we find evidence that genetic variants associated with increased HDL levels have a protective effect on clinically diagnosed Alzheimer's disease and vascular dementia risk, opposing the previously established risk-increasing effect[55,56]. Possible explanations for the observed opposing effect directions include Alzheimer's disease case definitions of previous reports relying on proxy cases and non-age-matched controls[57]. Our findings highlight that lipid metabolism pathways may influence risk of dementia and T2D in different directions. Further studies are needed to disentangle the relationship between HDL levels, T2D and dementia.

We identify evidence of potential causal effects of T2D genetic predisposition on asthma, depression, and COPD with opposing directions across global populations. These opposing directions of effect across different genetic ancestry groups may reflect differences in environmental factors, such as distinct effects of adiposity on T2D pathophysiology[58,59]. A further explanation may be the different strengths of association between the T2DGGI genetic clusters and the genetic ancestry groups. For example, in the latest T2DGGI work, it was

shown that variants assigned to the obesity cluster have greater allelic effects on T2D in EUR than EAS, whereas those assigned to the beta cell clusters have a greater allelic effect on T2D in EAS compared to EUR[11].

Modest effect sizes are inherent to MR analyses of complex diseases, reflecting their polygenic basis and the underlying liability threshold model[60]. Genetic variants generally have small effects on complex traits, limiting instrument strength and resulting in modest causal effect estimates that should not be directly equated with clinical risk. Nevertheless, these estimates provide valuable insights into potential causal mechanisms, supported here by comprehensive sensitivity analyses. Given the broad nature of the cardiometabolic traits used to cluster the T2DGGI genetic risk variants, they are posited to influence the risk for several diseases, including the comorbidities investigated here. We addressed the potential of confounding or mediation relationships by adjusting the MR estimates for the T2D-related cardiometabolic traits in a multivariable MR approach. If the adjusted effect of T2D genetic predisposition on comorbidity risk is attenuated to zero, care should be taken when interpreting the results given that the unadjusted estimates might be biased. Although derived from biological associations, one limitation of the eight T2D mechanistic clusters is that they reflect cardiometabolic processes, excluding other potential biological mechanisms involved in T2D. Moreover, the T2DGGI genetic risk variants were clustered based on a hard-clustering approach, which assigns all risk variants to exactly one cluster, with no overlap. The lack of outliers in this approach might decrease the robustness of each cluster.

The sample size of the T2DGGI GWAS meta-analysis used to identify T2DGGI genetic risk variants and create the mechanistic clusters is much larger than some of the non-cardiovascular comorbidity GWAS used in this work. This may influence the power to detect causal effects in the reverse MR analysis. Moreover, some of the GWAS datasets used in our analyses include phenotypes derived from self-reported data, which may introduce a degree of misclassification. While this is a common feature of large biobank studies, we note it as a potential limitation and encourage cautious interpretation of results for traits with less precise definitions. We acknowledge that validation of the findings using sex-specific T2D GWAS data is needed for sex-specific diseases. A well-known limiting factor in statistical genetics analyses is the relative paucity of GWAS data from diverse global populations[61]. In our work, this bias might lead to T2D genetic IVs having a different effect in under-represented global populations and, hence, being less powerful IVs for these populations. Despite the limitations, we identify evidence of a potential causal effect of genetic predisposition for T2D on rheumatoid arthritis only when meta-analyzing the causal estimates across global populations, highlighting the discovery power gain of integrating diverse GWAS data. Our results underscore the need for the community to continue to pursue efforts to further increase diversity in genetic studies. Finally, we note that future work leveraging high-resolution multi-omics data from primary tissues or experimental models will be important to further validate and mechanistically interpret the causal relationships proposed here.

In conclusion, we provide evidence of potential causal links between different mechanistic subtypes of T2D genetic predisposition and its non-cardiovascular comorbidities. These findings can inform preventive strategies to mitigate the onset of long-term co-occurring conditions by stratifying and monitoring patients based on their genetic burden to T2D mechanistic subtypes. Our work paves the way for more stratified treatment approaches aligned with the genetic and multimorbidity profiles of patients.

## Methods

This study used only publicly available summary statistics data. No new data were collected, and no institutional ethics approval was required. All data access complied with the relevant data use policies and ethical regulations of the original studies.

### Datasets

The T2DGGI consortium considered six genetic ancestry groups, which refer to the 1000 Genomes Project phase 3[39]: European, East Asian, African American, admixed American, South Asian, and South African (Supplementary Table 2). In the meta-analysis across all global populations (2,535,601 individuals including 428,452 cases), 1289 index ($r^2 < 0.05$) T2DGGI genetic risk variants were identified at genome-wide significance (p-value < $5 \times 10^{-8}$) (Supplementary Table 2).

Previous efforts from the T2DGGI consortium have clustered the 1289 T2DGGI genetic risk variants based on their association profile with 37 cardiometabolic traits. The traits used for clustering included glycaemic traits, anthropometric measures, body fat and adipose tissue volume, blood pressure, circulating plasma lipids levels and liver function and lipid metabolism biomarkers. A hard clustering approach was performed in an unsupervised manner. This resulted in eight non-overlapping mechanistic clusters of T2D risk variants that represent distinct biological pathways: obesity (n = 233), beta-cell associated with positive proinsulin (n = 91), beta-cell associated with negative proinsulin (n = 89), lipodystrophy (n = 45), liver and lipid metabolism (n = 3), residual glycaemic (n = 389), body fat (n = 273) and metabolic syndrome (n = 166). We used the T2DGGI GWAS summary statistics to perform MR analyses within genetic ancestry groups[11]. For all other diseases and quantitative traits employed in this work, an overview of the sample sizes and underlying populations is found in Table 1 and Supplementary Data 8. The T2D comorbidities were selected after a comprehensive literature review of observational studies and the availability of GWAS data. Despite its observational association with T2D, we have not included Parkinson's disease nor all-cause dementia to our study due to potential bias from proxy-case GWAS.

### Approaches to select instrumental variables for T2D

To best maintain the robustness of the T2DGGI genetic risk clusters, we used all 1289 T2D GWAS index risk variants identified in the T2DGGI global meta-analysis to derive the main results. Index variants were defined as genome-wide significant (p-value < $5 \times 10^{-8}$) variants with $r^2 < 0.05$ over a 5 Mb window[11]. If any genetic instrumental variable (IV) was not present in the outcome trait, we replaced it with an LD-based proxy ($r^2 > 0.8$) if available using the *LDlinkR::Ldproxy()* R function (v1.3.0)[62].

Using all T2D index risk variants, our definition of independent IVs is not as strict as the one employed by the *TwoSampleMR* R package (v0.5.7)[60], defined as LD-based clumped genome-wide significant (p-value < $5 \times 10^{-8}$) variants with $r^2 < 0.001$ over a 10 Mb window. To address this, we performed sensitivity analyses to select IVs for T2D. Firstly, we have removed all IVs with evidence of weak instrumental bias, defined as a per IV F-statistic = $(beta^2/se^2) < 10$, where *beta* is the effect size estimate, and *se* is its standard error from the T2D GWAS[63,64]. Secondly, we compared the effect magnitude of our results with alternative approaches to select T2D IVs. We have employed three additional approaches to define T2D IVs: selecting one variant per locus, LD-based clumping all the T2D index risk variants, and performing cluster-wise LD-based clumping (Supplementary Figs. 30–55, Supplementary Table 3, Supplementary Data 9).

### Selection of instrumental variables for other traits

For all other traits employed here (non-cardiovascular comorbidities and cardiometabolic traits), we defined IVs as LD-based clumped genome-wide significant (p-value < $5 \times 10^{-8}$) variants with $r^2 < 0.001$ over a 10 Mb window. Clumping was performed with PLINK (v2)[65] and LD was calculated based on the 1000 Genomes Project phase 3 release[39] with matching genetic ancestry groups. If any IV was absent in the T2D matching genetic ancestry group GWAS, we replaced it with a proxy. We used the output of PLINK, which assigns all variants to a clumped result, to search for proxies using the *ieugwasr::ld_matrix()* R function (v1.0.2). We removed variants with an F-statistic < 10[63].

## Two-sample Mendelian randomization analysis

Following the STROBE-MR guidelines[66], we performed bi-directional two-sample MR analyses[67] between genetic predisposition for T2D and 21 non-cardiovascular comorbidities. We used the *TwoSampleMR* R package (v0.5.7), which is curated by MR-Base[68]. It has been shown that the estimates of two-sample MR remain unbiased in the presence of sample overlap between exposure and outcome when using large sample sizes[69]. All analyses were performed within genetic ancestry groups using the LD panel from the corresponding genetic ancestry from the 1000 Genomes Project phase 3 release[39] to conduct clumping when necessary. Using matching genetic ancestry group data between exposure and outcome increases the accuracy of the causal estimate by reducing bias due to heterogeneous underlying data distributions.

For the main results, we applied the inverse variance weighted (IVW) method, which performs a random-effects meta-analysis of the Wald ratio estimate of each IV. If only one IV was available, the Wald ratio estimate was used. We applied additional MR methods to ensure consistency of causal effect direction under different methodological assumptions, namely the weighted median, MR-Egger[70], MR-PRESSO[71] and Steiger-filtered IVW methods[72]. We ran the MR-PRESSO distortion test with 1000 iterations to compute the null distribution for the outlier test and increased it to 1500 if it failed. If the distortion test was significant ($p$-value < 0.05), we considered the effect estimate of the MR-PRESSO outlier-corrected method. Otherwise, we used the estimate of the raw MR-PRESSO result. To facilitate interpretability when using binary traits as exposure, we report the MR results as OR for the outcome per doubling (2-fold change) in genetically determined dichotomous exposure risk. This unit is calculated by multiplying the MR effect sizes by ln(2) and converting them to OR via exponentiation[73].

## Assessment of the Mendelian randomization assumptions

MR relies on three main assumptions: relevance, independence and the exclusion restriction criteria (Supplementary Note 1, Supplementary Table 5). The relevance assumption states that the IVs need to be strongly associated with the exposure and can be directly tested by selecting genome-wide significant and strong (F-statistic > 10) IVs. The independence assumption ensures that the IVs are not associated with any confounder of the exposure-outcome association. The exclusion restriction criterion assumes no horizontal pleiotropy by ensuring that the IVs affect the outcome only through the exposure and not through any alternative pathways. Both assumptions cannot be directly tested but can be assessed via sensitivity analyses. We have conducted multivariable MR analyses to address the independence assumption by adjusting the univariable MR effect for known biological confounders or mediators of the exposure and outcome relationship. To test for horizontal pleiotropy, we applied the MR-Egger intercept test (*TwoSampleMR::mr_pleiotropy_test()*) and the MR-PRESSO outlier and distortion test. Both methods provide causal estimates corrected for the detected pleiotropy, and we looked for a concordant effect direction across different MR methods and the IVW estimate. Moreover, we assessed heterogeneity using $I^2$, a measure based on Cochran's Q-statistic (*TwoSampleMR::mr_heterogeneity()*) that is more interpretable and independent of the number of studies[74]. Evidence for heterogeneity, defined as $I^2 > 50$, implies that some IVs may influence the outcome through pathways other than the exposure, possibly violating the exclusion restriction assumption. Finally, we have tested for reverse causation by applying a bi-directional MR framework and a directionality test based on the Steiger filter (*TwoSampleMR::directionality_test()*) (Supplementary Data 10). Finally, we compared our results with the MR-Clust method, which groups together IVs with similar causal effect estimates on the outcome trait into distinct cluster[22] (Supplementary Note 3, Supplementary Data 11).

## Cluster-stratified two-sample Mendelian randomization analysis

To infer the causal effects of the eight mechanistic clusters of T2D genetic risk from the T2DGGI GWAS meta-analysis on the analyzed non-cardiovascular comorbidities, we conducted additional MR analyses restricting the T2D IVs to risk variants assigned to each cluster. For these analyses, we followed the same sensitivity analysis procedures described above. In addition, we used the correlated IVW method implemented in the *MendelianRandomization* R package (v0.10)[75], which allows for correlated IVs. This method is designed for less than 500 IVs and, therefore, could not be employed in the MR analyses using all 1289 T2DGGI genetic risk variants. As a sensitivity analysis to assess the specificity of the clusters, we performed a leave-one-out cluster MR analysis using as IVs the T2DGGI genetic risk variants, excluding all variants assigned to one cluster at a time.

## Meta-analysis across genetic ancestry groups

In cases of multiple publicly available GWAS summary statistics for a comorbidity from different genetic ancestry groups, we conducted a meta-analysis of the MR estimates across these groups. We used the *rma.uni()* function of the *metafor* R package (v4.6)[76] to conduct a random-effects meta-analysis using the IVW estimate. We tested for heterogeneity using a restricted maximum likelihood estimator (REML). We considered evidence for heterogeneity if $I^2 > 50$.

## Definition of significance

We account for multiple testing burden for both tested directions (T2D genetic predisposition on comorbidity risk and genetic predisposition for comorbidity on T2D risk) together by correcting the $p$-values of the IVW estimate and, if applicable, the correlated IVW estimate across all analyses (including the meta-analyses) using the FDR method (referred to as $q$-values). We defined statistical significance as IVW estimates and, if applicable, correlated IVW estimates with a $q$-value < 0.05.

## Comparison with previous MR studies

We compare our causal estimates with previous MR studies looking at the investigated disease pairs (Supplementary Fig. 1, Supplementary Table 1)[12,13,15–20,24–33]. We considered only results from the IVW method. To match the scale reported in our study, we multiplied the MR effect sizes from previous studies by ln(2) and converted them to OR via exponentiation. We have not compared the potentially different approaches to select IVs nor the criteria to define disease cases employed in the respective GWAS. For anorexia nervosa, vascular dementia and OCD, we did not find MR studies using T2D genetic predisposition as exposure.

## Multivariable Mendelian randomization analysis with cardiometabolic traits

For the cluster-stratified genetic predisposition for T2D MR results with FDR < 5%, we subsequently performed multivariable MR (MVMR) analyses using the mechanistic clusters of T2D genetic predisposition and several cardiometabolic traits as exposure and the non-cardiovascular comorbidities as outcome (Supplementary Figs. 5–28). By adjusting cluster-stratified MR estimates for cardiometabolic traits causally associated with the comorbidities, we can estimate the direct effect of the mechanistic clusters of T2D genetic predisposition on its comorbidities. To compare the univariable and multivariable MR results, we first performed univariable MR analyses using the cardiometabolic traits as exposures and the investigated T2D comorbidities as outcomes. The results were adjusted for multiple testing using FDR correction. IVs were selected using the same approach as for the comorbidities. We then conducted an MVMR analysis only for the cardiometabolic traits that showed a potential causal effect on comorbidity at FDR of 5%. We applied the *TwoSampleMR::mv_multiple()* function, which uses the IVW method to perform the MVMR analysis. We performed sensitivity analyses using the *MVMR*

R package (v0.4)[77], including testing for heterogeneity (*qhet_mvmr()*), pleiotropy (*pleiotropy_mvmr()*) and instrumental strength of each exposure (*strength_mvmr()*).

### Phenome-wide association study of cluster-stratified T2D risk variants

To identify non-cardiovascular phenotypes associated with the T2DGGI clusters, we performed PheWAS in each cluster of genetic predisposition for T2D using the National Institutes of Health (NIH) *All of Us* cohort[34], which was not part of the T2DGGI GWAS meta-analysis nor any of the comorbidities GWAS. All research was carried out on the *All of Us* Researcher Workbench. Quality control (QC) and genetic ancestry group assignment were performed on 206,000 participants with EHR and WGS data. We pruned the dataset to a maximal set of unrelated individuals using a kinship coefficient <0.1 and removed samples with ambiguous sex (Supplementary Table 4). To reduce the computational complexity, we used the *All of Us* ACAF variant set, which retains those variants with an allele frequency (AF) > 1% or allele count (AC) > 100 in any computed genetic ancestry group.

For each genetic ancestry group, we calculated the first 10 principal components (PCs) using smartpca (v7.2.1)[78] with LD-pruned genotypes and the *fastmode* option enabled. To derive the LD-pruned genotypes, we restricted variants to autosomal variants that were present in 1000 Genomes Project phase 3 release[39], not in the major histocompatibility complex (MHC) region, minor allele frequency (MAF) ≥ 1%, and Hardy-Weinberg equilibrium (HWE) $p$-value ≥ 1×10$^{-6}$. LD pruning was conducted using PLINK (v2)[65] with $r^2$ = 0.05 in a 1000 KB window with an 80b step size. For each genetic ancestry group and T2DGGI genetic cluster, we calculated a GRS summing the cluster-stratified risk alleles, weighted by the effect sizes from each genetic ancestry group. The GRS was adjusted for the 10 first PCs and scaled within each genetic ancestry group.

For each genetic ancestry group and T2DGGI genetic cluster, we performed a PheWAS using the PheTK package (v0.2.1rc5)[79]. Phenotype code (phecode) counts were calculated using the count_phecode() function of PheTK, with the following options: 'phecode_version = "X"' and 'icd_version = "US"'. Participants without any recorded phecodes were excluded to avoid incomplete EHRs. PheWAS were conducted with the PheWAS function of PheTK using logistic regression and the model 'phecode ~ sex + age at last EHR + PCs 1-10 + cluster GRS'. Only phecodes with a minimum case count of 50 were included, and cases were defined as having at least two instances of the respective phecode. We considered phecodes from the following categories: mental, musculoskeletal, neurological, respiratory, and sense organs. Multiple testing correction was applied for each genetic ancestry group separately using a $q$-value < 0.05. We meta-analyzed the cluster-stratified PheWAS across genetic ancestry groups using the random-effects IVW method and the *rma.uni()* function of the metafor R package (v4.6)[76]. The meta-analysis results were FDR corrected for multiple testing separately.

### Reporting summary

Further information on research design is available in the Nature Portfolio Reporting Summary linked to this article.

## Data availability

Researchers can apply to access the individual-level data of the *All of Us* Research Program (https://researchallofus.org/) used to perform the PheWAS. The publicly available GWAS summary statistics used in this work are referenced below and in Supplementary Data 8. Alzheimer's disease: GWAS data has been deposited in The National Institute on Aging Genetics of Alzheimer's Disease Data Storage Site (NIAGADS)—a NIA/NIH-sanctioned qualified-access data repository, under accession NG00075. Anorexia: https://pgc.unc.edu/for-researchers/download-results/. Asthma: https://www.globalbiobankmeta.org/resources.

ADHD: https://pgc.unc.edu/for-researchers/download-results/. Autism: https://pgc.unc.edu/for-researchers/download-results/. Back pain: the dataset can also be accessed under 'Chronic back pain' from https://gwasarchive.org. Bipolar disorder: https://pgc.unc.edu/for-researchers/download-results/. CTS: https://www.decode.com/summarydata/. Cataracts: https://www.ebi.ac.uk/gwas/studies/GCST90014268. COPD: https://www.globalbiobankmeta.org/resources. Depression EUR: https://pgc.unc.edu/for-researchers/download-results/. Depression non-EUR: https://pgc.unc.edu/for-researchers/download-results/. Epilepsy: https://www.ebi.ac.uk/gwas/studies/GCST90271608. Erectile dysfunction: http://www.geenivaramu.ee/tools/ED_AJHG_Bovijn_et_al_2018.gz. Glaucoma: https://xikunhan.github.io/site/publication/. OCD: https://pgc.unc.edu/for-researchers/download-results/. Osteoarthritis: https://msk.hugeamp.org/downloads.html. Osteoporosis: https://pheweb.jp/. PCOS: https://doi.org/10.17863/CAM.27720. Rheumatoid arthritis: https://www.ebi.ac.uk/gwas/studies/GCST90132223. Schizophrenia: https://pgc.unc.edu/for-researchers/download-results/.

## Code availability

The code used to perform all the MR-related and PheWAS analyses is publicly available and archived on Zenodo: https://doi.org/10.5281/zenodo.15168490. *LDlinkR* R package (v1.3.0) https://doi.org/10.32614/CRAN.package.LDlinkR. *TwoSampleMR* R package (v0.5.7) https://mrcieu.github.io/TwoSampleMR/index.html. PLINK (v2) https://www.cog-genomics.org/plink/2.0/. MR-PRESSO https://doi.org/10.1038/s41588-018-0099-7. *MendelianRandomization* R package (v0.10) https://doi.org/10.32614/CRAN.package.MendelianRandomization. *metafor* R package (v4.6) https://doi.org/10.32614/CRAN.package.metafor. *MVMR* R package (v0.4) https://wspiller.github.io/MVMR/. smartpca (v7.2.1) https://christianhuber.github.io/smartsnp. PheTK package (v0.2.1rc5) https://pypi.org/project/PheTK/. *ieugwasr* R function (v1.0.2) https://doi.org/10.32614/CRAN.package.ieugwasr.

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

## Acknowledgements

O.B. has received funding from the European Union's Horizon 2020 research and innovation program under Grant Agreement No 101017802 (OPTOMICS). Supported in part by the National Center for Advancing Translational Sciences, CTSI grant UL1TR001881 and the National Institute of Diabetes and Digestive and Kidney Disease Diabetes Research Center (DRC) grant DK063491 to the Southern California Diabetes Endocrinology Research Center. Infrastructure for the CHARGE Consortium is supported in part by the National Heart, Lung and Blood Institute (NHLBI) grant R01HL105756. J.M.M. is supported by American Diabetes Association grant #11-22-ICTSPM-16 and by NHGRI U01HG011723, by the National Institute Of Diabetes And Digestive And Kidney Diseases of the National Institutes of Health under Award Number R01DK137993 and U01 DK140757, AMP CMD award from RFP 6 from the Foundation for the National Institutes of Health, and a Medical University of Bialystok (MUB) grant from the Ministry of Science and Higher Education (Poland). C.N.S. is supported by American Diabetes Association grant #11-22-JDFPM-06 and the National Institute of Diabetes and Digestive and Kidney Diseases of the National Institutues of Health under award numbers R01DK118011 and R01DK136671. JBM was supported by UM1 DK078616. APM was supported by the NIHR Manchester Biomedical Research Centre (NIHR203308).

## Author contributions

A.L.A., O.B., H.T. and D.C. contributed equally to this project. B.F.V., A.P.M. and E.Z. jointly supervised this work. A.L.A., O.B., B.F.V., A.P.M. and E.Z. conceived the project and analysis plan. H.T., D.C., S.Y., X.Y., C.Z., J.C., A.C.W., K.S., J.M.M., C.N.S., J.B.M., M.V., G.D.S. and J.I.R. reviewed the analysis plan. A.L.A., O.B., H.T. and D.C. conducted all the analyses. The interpretation of the results was conducted by all authors. A.L.A. and O.B. drafted the manuscript, which was reviewed by all authors.

## Funding

## Competing interests

The authors have no competing interests.

## Additional information

[1]Institute of Translational Genomics, Helmholtz Zentrum München – German Research Center for Environmental Health, Neuherberg, Germany. [2]Medical Research Council (MRC) Epidemiology Unit, University of Cambridge, Cambridge, UK. [3]Univ Brest, Inserm, EFS, UMR 1078, Brest, France. [4]Center for Precision Health Research, National Human Genome Research Institute, National Institutes of Health, Bethesda, MD, USA. [5]British Heart Foundation Cardiovascular Epidemiology Unit, Department of Public Health and Primary Care, University of Cambridge, Cambridge, UK. [6]Heart and Lung Research Institute, University of Cambridge, Cambridge, UK. [7]Nevada Institute of Personalized Medicine, University of Nevada, Las Vegas, Las Vegas, NV, USA. [8]Programs in Metabolism and Medical and Population Genetics, Broad Institute of MIT and Harvard, Cambridge, MA, USA. [9]Center for Genomic Medicine, Kyoto University Graduate School of Medicine, Kyoto, Japan. [10]McGill Genome Centre, McGill University, Montreal, QC, Canada. [11]Department of Epidemiology, School of Public Health, Nanjing Medical University, Nanjing, China. [12]Department of Biostatistics and Center for Statistical Genetics, University of Michigan, Ann Arbor, MI, USA. [13]Department of Biostatistics and Epidemiology, University of Massachusetts Amherst, Amherst, MA, USA. [14]USDA/ARS Children's Nutrition Center, Baylor College of Medicine, Houston, TX, USA. [15]Department of Diabetes and Metabolic Diseases, Graduate School of Medicine, University of Tokyo, Tokyo, Japan. [16]Diabetes Unit and Center for Genomic Medicine, Massachusetts General Hospital, Boston, MA, USA. [17]Harvard Medical School, Boston, MA, USA. [18]Department of Medicine, Harvard Medical School, Boston, MA, USA. [19]Division of General Internal Medicine, Massachusetts General Hospital, Boston, MA, USA. [20]Corporal Michael J. Crescenz VA Medical Center, Philadelphia, PA, USA. [21]Department of Genetics, University of Pennsylvania Perelman School of Medicine, Philadelphia, PA, USA. [22]Department of Biostatistics, Epidemiology and Informatics, University of Pennsylvania Perelman School of Medicine, Philadelphia, PA, USA. [23]MRC Integrative Epidemiology Unit, University of Bristol, Bristol, UK. [24]Institute for Translational Genomics and Population Sciences, Department of Pediatrics, Lundquist Institute for Biomedical Innovation at Harbor-UCLA Medical Center, Torrance, CA, USA. [25]Department of Systems Pharmacology and Translational Therapeutics, University of Pennsylvania Perelman School of Medicine, Philadelphia, PA, USA. [26]Institute for Translational Medicine and

Therapeutics, University of Pennsylvania Perelman School of Medicine, Philadelphia, PA, USA. [27]Centre for Genetics and Genomics Versus Arthritis, Centre for Musculoskeletal Research, The University of Manchester, Manchester, United Kingdom. [28]NIHR Manchester Biomedical Research Centre, Manchester University NHS Foundation Trust, Manchester Academic Health Science Centre, Manchester, UK. [29]TUM School of Medicine and Health, Technical University of Munich University Hospital, Munich, Germany. [30]These authors contributed equally: Ana Luiza Arruda, Ozvan Bocher, Henry J. Taylor, Davis Cammann. [31]These authors jointly supervised this work: Benjamin F. Voight, Andrew P. Morris, Eleftheria Zeggini. ✉e-mail: analuiza.arruda@helmholtz-munich.de; eleftheria.zeggini@helmholtz-munich.de

