## [Transparent Peer Review file · Nature Communications]

The effect of type 2 diabetes genetic predisposition on non-cardiovascular comorbidities

Corresponding Author: Professor Eleftheria Zeggini

Version 0:

Reviewer comments:

Reviewer #1

(Remarks to the Author)

This manuscript presents a study with an unconventional and somewhat disjointed structure, comprising two distinct and insufficiently integrated components: one focusing on Mendelian Randomization (MR) and the other on metformin. The objectives and key outcomes, as stated in the abstract, are relatively unclear:

“When surveyed in populations across the globe, we observe opposing effect directions for depression, asthma and chronic obstructive pulmonary disease between populations. We identify a putative causal link between T2D genetic predisposition and osteoarthritis. To underscore the translational potential of our findings, we intersect high-confidence effector genes for osteoarthritis with targets of T2D-approved drugs and identify metformin as a potential candidate for drug repurposing in osteoarthritis.”

The focus on osteoarthritis is poorly justified and the last sentence not supported by any data.

This abstract highlights one of the main concerns of the reviewer: the study presents a broad and somewhat fragmented set of analyses, with findings in many directions, often characterized by modest effect sizes and, in some cases, only marginal p-values—an issue commonly encountered in MR studies.

The Introduction largely focuses on epidemiological trends and GWAS findings, but forgot to introduce the rationale for using MR or to address the issue of causal inference. There is no meaningful framing of this study’s methodological approach and its putative novelty in the context of important existing MR literature.

In the Discussion, MR literature is addressed only briefly, with a single sentence and a few generic references, lacking any substantive engagement with prior MR studies on type 2 diabetes (T2D) and its comorbidities. This omission is surprising, as numerous MR studies have already investigated relationships between T2D and conditions such as depression, various cancers, and other chronic diseases. The authors’ exploration of multiple T2D (and obesity) comorbidities would benefit significantly from contextualization within this existing body of work.

There is also a lack of clarity regarding the cohorts and datasets used. While the “All of Us” resource is mentioned a few times, it does not appear to be the primary dataset. The manuscript should more clearly specify the origin, characteristics, and reliability of the datasets and phenotypes used. This is especially important given that some phenotypes may be derived from self-reported questionnaires, which could introduce significant misclassification bias and weaken the reliability of the MR analyses.

The study is entirely statistics-driven, which in itself is not problematic, but it misses the opportunity to follow up on the most promising associations with molecular or mechanistic evidence. This is particularly evident in the section concerning metformin, which is insufficiently detailed. The source of the data, the rationale for its inclusion, and its implications are unclear. Without any downstream validation—whether through additional human omics data or experimental models—the translational relevance remains speculative at best. In its current form, the metformin component appears poorly integrated, inadequately supported, and difficult to interpret. For these reasons, the reviewer strongly recommends removing this section from the manuscript unless further data and clarification can be provided.

Reviewer #2

(Remarks to the Author)

- The abstract is written in a way that the reader is not entirely sure what to expect and is worded quite bizarrely. Please can

the authors look at other MR papers published in Nat Comms and use similar/more conventional language and structuring to those.

- It's probably more common to use the term 'observationally' associated rather than 'epidemiologically' associated
- 58-71 – abstract (and some of the body text) is bogged down in verbosity. I don't know if it's our place as reviewers to make this comment, but it should more plainly lay out the results.

- The word 'putative' is used 25 times in this paper – we think this is unnecessary, especially in the Abstract where it appears multiple times.

- Results:

o Paragraph 1 overview is a bit out of place here – that's more methods and detracts from the results

o Expressions such as 'power bracket' are a bit odd; this sounds electrical almost, please amend to more conventional language used in the field.

o Admixed Americans is quite a broad term and not a group that is represented in 1k genomes. Clarification is needed.

o In the section on different genetic ancestries lines 310 to 317 are discussion/interpretation, not results – please move this to the Discussion or take it out.

o Metformin results: we were a bit dubious about these results, as no numerical results/effect sizes are provided. The description of these results is a bit vague, and they appear to come out of nowhere with little rationale provided.

o 193 – consistent language – etiology/causality are thrown around haphazardly.

o 320 – figure 4 – we feel like this figure could do with being a 3x1 figure where the rows correspond to ancestries and the y-axis reflects the comorbidities.

- Methods:

o Instrument selection criteria: what's the justification for the commonly employed p-value, r² and distance thresholds?

There is no reference provided, and it is my understanding that there is no real consensus in the field as to how instruments should be selected.

o Can the authors provide a reference for the F-statistic method/calculation and >10 criterion?

o Datasets: it's not clear what table is 'Table_T2DGGI_samplesize', is this supp table 9? If so, the ancestry groups are not the same in the table as in the text. The table has a Hispanic group, but no admixed group and the text is the other way round. Please clarify this and refer to the correct table or amend the table, if not. Admixed and Hispanic aren't the same.

o 266 – 'evidence of putative causal links' -> 'indicating a multifactorial association'. 'suggestions of supposed causal links' which then 'indicate' such an association – the language throughout is too timid and hamstrings the impact of what is being said.

o Using Plink 1.9 also odd, do the results differ at all using the updated software, ie Plink 2.0?

o Any consideration given to analysing vascular and all-cause dementia?

- Discussion:

o Metformin findings: we think that the way this is written is a bit of a reach re saying that the authors have identified it as a potential repurposing opportunity for osteoarthritis. This needs softening and more details of the actual results, as highlighted above. This also needs to be triangulated and replicated with other approaches, as MR is not the only method used to ascertain causality.

o 412: the authors declare that they were limited by power, but at no point have they conducted formal power analyses (there are calculators for this for MR studies) and looking at the number of cases in the T2DGGI GWAS they look reasonable... perhaps they can tone this down a little, as they don't know for sure that they were underpowered?

o We think a bit more discussion is needed around certain results, e.g. Alzheimer's protective for diabetes...

Reviewer #3

(Remarks to the Author)

Version 1:

Reviewer comments:

Reviewer #1

(Remarks to the Author)

This thanks the authors to have understood reviewers' comments and followed most of their suggestions. The paper is much improved and in my opinion acceptable for publication. The metformin study should be continued until completion with more physiology for publication somewhere.

Reviewer #2

(Remarks to the Author)

Thank you to the authors for substantially revising their manuscript based on our suggestions. Final things to please amend are:

1) 'get attenuated' (change to something like 'become attenuated') which sounds very casual;

2) 'Despite being observationally associated with T2D' - informal, please amend to something like 'despite its observational

association'

3) Please split this very long and wordy sentence into two: This is further supported by the attenuation of the identified effect after adjustment for genetic predisposition for higher HDL levels, which suggests that the effect of genetic predisposition for both dementia types on T2D risk is at least partially explained by HDL levels.

Reviewer's Comments:

Reviewer #1 (Remarks to the Author)

This manuscript presents a study with an unconventional and somewhat disjointed structure, comprising two distinct and insufficiently integrated components: one focusing on Mendelian Randomization (MR) and the other on metformin.

We thank the reviewer for their comments that have helped to substantially improve the presentation of our work. Following the reviewer's explicit recommendation, we have decided to remove the metformin part of our manuscript.

The objectives and key outcomes, as stated in the abstract, are relatively unclear:

"When surveyed in populations across the globe, we observe opposing effect directions for depression, asthma and chronic obstructive pulmonary disease between populations. We identify a putative causal link between T2D genetic predisposition and osteoarthritis. To underscore the translational potential of our findings, we intersect high-confidence effector genes for osteoarthritis with targets of T2D-approved drugs and identify metformin as a potential candidate for drug repurposing in osteoarthritis." The focus on osteoarthritis is poorly justified and the last sentence not supported by any data.

We understand the reviewer's concerns and have carefully reviewed the abstract as well as the structure of the manuscript to increase the understanding of the motive and results of our analyses. We have now removed the metformin paragraph, as suggested by the reviewer.

New abstract:

"Type 2 diabetes (T2D) is associated with a wide range of non-cardiovascular non-oncologic comorbidities. To move beyond associations and evaluate potential causal effects between T2D genetic predisposition and 21 comorbidities, we apply Mendelian randomization analysis using genome-wide association study (GWAS) results across multiple genetic ancestries. In addition, leveraging eight non-overlapping mechanistic clusters of T2D genetic profiles, each representing distinct biological pathways, we investigate causal links between cluster-stratified T2D genetic predisposition and risk of comorbidities. We identify causal effects of T2D genetic predisposition driven by distinct genetic clusters. For example, the risk-increasing effects of T2D genetic predisposition on cataracts and erectile dysfunction are primarily attributed to adiposity and glucose regulation mechanisms, respectively. We observe opposing effect directions across different genetic ancestries for depression, asthma and chronic obstructive pulmonary disease. Our findings leverage the heterogeneity underpinning T2D genetic predisposition to prioritize biological mechanisms underlying causal relationships with different comorbidities."

This abstract highlights one of the main concerns of the reviewer: the study presents a broad and somewhat fragmented set of analyses, with findings in many directions,

often characterized by **modest effect sizes and, in some cases, only marginal p-values—an issue commonly encountered in MR studies.**

Modest effect sizes are indeed expectable in MR studies of complex diseases, which are based on a liability model. Due to the polygenic architecture of complex diseases, genetic variants identified in GWAS have usually modest effects on their risk, and so are not necessarily strong predictors of the exposure. We added these considerations to the discussion, as well as emphasized that effect sizes in MR cannot be easily translated to the clinic. Regarding the set of analyses performed, we followed the STROBE guideline and an overview of the different sensitivity analyses conducted is presented in the Supplementary Note.

Addition to the Discussion section:

“Modest effect sizes are inherent to MR analyses of complex diseases, reflecting their polygenic basis and the underlying liability threshold model[81]. Genetic variants generally have small effects on complex traits, limiting instrument strength and resulting in modest causal effect estimates that should not be directly equated with clinical risk. Nevertheless, these estimates provide valuable insights into potential causal mechanisms, supported here by comprehensive sensitivity analyses.”

The Introduction largely focuses on epidemiological trends and GWAS findings, but forgot to introduce the rationale for using MR or to address the issue of causal inference.

We agree with the reviewer that an early introduction to the methodology and its challenges in our manuscript would be beneficial to the readers. In the updated version of our manuscript, we have introduced the idea behind MR and discussed the challenges of causal inference, including pleiotropy and unknown confounders.

Addition to the Introduction section:

“To move beyond associations, we evaluate causal relationships between T2D genetic predisposition and 21 non-cardiovascular non-oncologic comorbidities by employing bi-directional Mendelian randomization (MR). MR is a causal inference method that uses genetic variants as instrumental variables (IVs) to estimate the causal effect of an exposure on an outcome, leveraging the random allocation of alleles at conception to reduce confounding and reverse causation. For MR estimates to be valid, three core assumptions must hold: (1) the IVs are robustly associated with the exposure of interest; (2) the IVs are not associated with any confounders of the exposure-outcome relationship; and (3) the IVs influence the outcome solely through the exposure, not via alternative pathways (no horizontal pleiotropy). We acknowledge that violations of these assumptions pose challenges to causal inference and have implemented sensitivity analyses in accordance to the STROBE guidelines to test the robustness of our findings (Methods, SupNote).”

There is no meaningful framing of this study’s methodological approach and its putative novelty in the context of important existing MR literature.

Following the reviewer's suggestion, we have moved the overview of our study design to the Introduction section, where we now explain our biologically-meaningful cluster-stratified MR approach and place it under the context of current MR literature. In addition, we have highlighted how our method considers different T2D genetic profiles, recognizing the heterogeneous nature of genetic predisposition for this disease.

Addition to the Introduction section

"Cluster-stratified MR analysis has been previously employed to identify specific biological mechanisms driving causal relationships and potential pleiotropic pathways[21-23]. Here, in addition to performing pairwise bi-directional MR analyses, we investigated cluster-stratified effects of T2D genetic predisposition on its comorbidities by restricting the IVs to T2DGGI genetic risk variants assigned to each mechanistic cluster identified in the latest T2DGGI GWAS meta-analysis. Our biologically informed cluster-stratified MR approach aims to identify distinct T2D genetic profiles causally associated with 21 non-cardiovascular diseases, spanning five classes: musculoskeletal, respiratory, reproductive, neuropsychiatric, and ophthalmic conditions. The different T2D genetic profiles highlight potential biological pathways underlying specific comorbidity pairs."

In the Discussion, MR literature is addressed only briefly, with a single sentence and a few generic references, lacking any substantive engagement with prior MR studies on type 2 diabetes (T2D) and its comorbidities. This omission is surprising, as numerous MR studies have already investigated relationships between T2D and conditions such as depression, various cancers, and other chronic diseases. The authors' exploration of multiple T2D (and obesity) comorbidities would benefit significantly from contextualization within this existing body of work.

We thank the reviewer for this important point. We have now compiled a new supplemental figure (see below) directly comparing our causal estimates with those from previous MR studies, allowing the reader to contextualize our results within the broader literature. As our manuscript centers on non-cardiometabolic and non-oncologic comorbidities of T2D, we do not systematically review prior MR studies on cancer, but we agree this is a rich and active area of research and have highlighted the scope of our study ("non-cardiovascular non-oncologic comorbidities") in the abstract.

Addition to the Results section:

"When comparing our causal estimates with results of previous MR studies investigating the effect of T2D genetic predisposition on disease risk, we find mostly consistent directions of effect (Supplemental Figure 1, SupTab2)[12, 13, 15-20, 47-56]. For anorexia nervosa, vascular dementia and OCD, we did not find MR studies using T2D genetic predisposition as the exposure for comparison."

Supplemental Figure 1: Comparison of ours and previous studies results of two-sample Mendelian randomization (MR) analysis of genetic predisposition for type 2 diabetes (T2D) on non-cardiovascular comorbidity risk for the causal relationships. Causal estimates are expressed as the odds ratio (OR) for each comorbidity per doubling (2-fold increase) in genetically determined dichotomous T2D risk. (CI = confidence interval; CTS = Carpal tunnel syndrome; ADHD = attention-deficit/hyperactivity disorder; OCD = obsessive-compulsive disorder; PCOS = polycystic ovary syndrome).

Addition to the Methods section:

“We compare our causal estimates with previous MR studies looking at the investigated disease pairs (Supplemental Figure 1, SupTab2)[12, 13, 15-20, 47-56]. We considered only results from the IVW method. To match the scale reported in our study, we multiplied the MR effect sizes from previous studies by $\ln(2)$ and converted them to OR via exponentiation. We have not compared the potentially different approaches to select IVs nor the criteria to define disease cases employed in the respective GWAS. For anorexia nervosa, vascular dementia and OCD, we did not find MR studies using T2D genetic predisposition as exposure.”

Addition to the Discussion section:

“Despite potential differences in phenotype definition and selection of IVs, our genetic findings align with previous MR studies investigating the causal effect of overall genetic predisposition for T2D on comorbidity risk[12, 13, 15-20, 47-56] (Supplemental Figure 1).”

There is also a lack of clarity regarding the cohorts and datasets used. While the “All of Us” resource is mentioned a few times, it does not appear to be the primary dataset. The manuscript should more clearly specify the origin, characteristics, and reliability of the datasets and phenotypes used. This is especially important given that some phenotypes may be derived from self-reported questionnaires, which could introduce significant misclassification bias and weaken the reliability of the MR analyses.

We agree with the reviewer that more detail in the main text on the data used in our study is beneficial for the readers. We have emphasized the selection criteria for the GWAS included in the MR analyses in our Results section:

“The T2D comorbidities were selected after a comprehensive literature review of observational studies and the availability of GWAS summary statistics without the inclusion of proxy cases.”

To further improve clarity and readability, we have now added a table to the Results section with an overview of each GWAS dataset used in our MR analyses (Table 1), which includes genetic ancestry, sample size, case definitions, and the respective publication reference for each disease GWAS.

Disease	Genetic ancestry	N	Ncases	Ncontrols	Cases definition		Ref.
					SR	CD	
Alzheimer's disease	EUR	63936	21982	41954		x	[24]
Anorexia nervosa	EUR	72517	16992	55525	x	x	[25]
Asthma	EUR	1376071	121940	1254131	x	x	[26]
	EAS	341204	18549	322655	x	x	[26]
	AMR	18173	4069	14104	x	x	[26]
	SAS	27091	4015	23076	x	x	[26]
	AFR	32658	5051	27607	x	x	[26]
ADHD	EUR	225534	38691	186843		x	[27]

Autism	EUR	46350	18381	27969		X	[28]
Back pain	EUR	158025	29531	128494	X	X	[29]
Bipolar disorder	EUR	413466	41917	371549		X	[30]
CTS	EUR	1239680	48843	1190837		X	[31]
Cataracts	EUR	585243	67844	517399		X	[32]
COPD	EUR	995917	58559	937358	X	X	[33]
	AFR	29682	1978	27704	X	X	[33]
	AMR	15086	1503	13583	X	X	[33]
	EAS	329733	19044	310689	X	X	[33]
Depression	EUR	404529	170756	233773	X	X	[34]
	EAS	382936	21980	360956	X	X	[35]
	AMR	377959	25013	352946	X	X	[35]
	SAS	31681	4505	27176	X	X	[35]
	AFR	198497	36818	161679	X	X	[35]
Epilepsy	EUR	82482	29,944	52538		X	[36]
Erectile dysfunction	EUR	223805	6175	217630	X	X	[37]
Glaucoma	EUR	379422	29241	350181	X	X	[38]
OCD	EUR	9725	2688	7037		X	[39]
Osteoarthritis	EUR	826690	177517	649173	X	X	[40]
Osteoporosis	EUR	488501	8520	479981		X	[41]
	EAS	178726	9794	168932		X	[41]
PCOS	EUR	120023	5209	114814		X	[42]
Rheumatoid arthritis	EUR	97173	22350	74823		X	[43]
	EAS	173633	11025	162608		X	[43]
Schizophrenia	EUR	130644	53386	77258		X	[44]
	EAS	30761	14004	16757		X	[44]
	AFR	10070	6152	3918		X	[44]
	AMR	4324	1234	3090		X	[44]
Vascular dementia	EUR	466606	3892	462714		X	[45]

Table 1: Overview of non-cardiometabolic type 2 diabetes comorbidities GWAS, including genetic ancestry defined based on the continental super populations of the 1000 Genomes Project phase 3[44] (AMR = Admixed American; AFR = African; EAS = East Asian; EUR = European; SAS = South Asia), total sample size (N), number of cases (Ncases), number of controls (Ncontrols), case definition (SR = self-reported; CD = clinically diagnosed) and the corresponding GWAS publication. (ADHD = attention-deficit/hyperactivity disorder; COPD = chronic obstructive pulmonary disease; CTS = Carpal tunnel syndrome; OCD = obsessive-compulsive disorder; PCOS = polycystic ovary syndrome).

We used data from the All of Us Research Program specifically for the PheWAS analysis exploring associations between T2DGGI genetic clusters and its comorbidities. These results are independent of our MR analysis, since none of the MR GWAS data included the All Of Us resource. We now state clearly that the genetic variants and genetic clusters used throughout our analyses come from the T2DGGI

GWAS meta-analysis and have revised the Results section to clearly distinguish between the datasets used for PheWAS and those used for MR:

“To validate the MR findings, we performed a phenome-wide association study (PheWAS) for each T2DGGI genetic cluster using data from the All of Us Research Program, an ancestrally and culturally diverse cohort that was not included in the T2DGGI GWAS meta-analysis nor in any of the T2D comorbidity GWAS[57] (Methods).”

We also now explicitly note in the Discussion section that some of the GWAS used in our study may include phenotypes derived from self-reported data, which can introduce misclassification bias. We comment on this as a limitation and caution against overinterpretation of findings that may be affected by this source of measurement error.

“Moreover, some of the GWAS datasets used in our analyses include phenotypes derived from self-reported data, which may introduce a degree of misclassification. While this is a common feature of large biobank studies, we note it as a potential limitation and encourage cautious interpretation of results for traits with less precise definitions.”

The study is entirely statistics-driven, which in itself is not problematic, but it misses the opportunity to follow up on the most promising associations with molecular or mechanistic evidence. This is particularly evident in the section concerning metformin, which is insufficiently detailed. The source of the data, the rationale for its inclusion, and its implications are unclear. Without any downstream validation—whether through additional human omics data or experimental models—the translational relevance remains speculative at best. In its current form, the metformin component appears poorly integrated, inadequately supported, and difficult to interpret. For these reasons, the reviewer strongly recommends removing this section from the manuscript unless further data and clarification can be provided.

We thank the reviewer for their constructive feedback. In line with their suggestion, we have removed the metformin section from the revised manuscript.

Our work is designed as a hypothesis-generating investigation grounded in large-scale statistical genomics, aimed at identifying disease areas where T2D liability may have broad causal relevance. We have emphasized the goal of our study in the Discussion section:

“Here, we sought to identify non-cardiovascular, non-oncologic diseases causally affected by T2D genetic predisposition, and investigated the biological mechanisms underlying the identified causal effects.”

While we of course recognize the value of integrating mechanistic insights and molecular follow-up, this often requires data at a level of resolution such as gene expression, epigenetics, or proteomics in disease-relevant tissues that is currently

unavailable or insufficiently powered for many of the comorbidities we explore. We have added this consideration to the revised Discussion section:

“Finally, we note that future work leveraging high-resolution multi-omics data from primary tissues or experimental models will be important to further validate and mechanistically interpret the causal relationships proposed here.”

Reviewer #2 (Remarks to the Author)

- The abstract is written in a way that the reader is not entirely sure what to expect and is worded quite bizarrely. Please can the authors look at other MR papers published in Nat Comms and use similar/more conventional language and structuring to those. We thank the reviewers for their constructive feedback. We have rewritten the abstract as well as the rest of the manuscript using a more direct and MR-conventional language.

New abstract:

“Type 2 diabetes (T2D) is associated with a wide range of non-cardiovascular non-oncologic comorbidities. To move beyond associations and evaluate potential causal effects between T2D genetic predisposition and 21 comorbidities, we apply Mendelian randomization analysis using genome-wide association study (GWAS) results across multiple genetic ancestries. In addition, leveraging eight non-overlapping mechanistic clusters of T2D genetic profiles, each representing distinct biological pathways, we investigate causal links between cluster-stratified T2D genetic predisposition and risk of comorbidities. We identify causal effects of T2D genetic predisposition driven by distinct genetic clusters. For example, the risk-increasing effects of T2D genetic predisposition on cataracts and erectile dysfunction are primarily attributed to adiposity and glucose regulation mechanisms, respectively. We observe opposing effect directions across different genetic ancestries for depression, asthma and chronic obstructive pulmonary disease. Our findings leverage the heterogeneity underpinning T2D genetic predisposition to prioritize biological mechanisms underlying causal relationships with different comorbidities.”

- It's probably more common to use the term 'observationally' associated rather than 'epidemiologically' associated

We agree with the reviewers and have updated our manuscript accordingly.

- 58-71 – abstract (and some of the body text) is bogged down in verbosity. I don't know if it's our place as reviewers to make this comment, but it should more plainly lay out the results.

We appreciate the reviewers' constructive feedback. We have revised the abstract, Results and Discussion sections to improve clarity, ensuring the results are presented more directly and accessibly. Specifically, we have reduced the use of the words *putative* and *potential*.

- The word 'putative' is used 25 times in this paper – we think this is unnecessary, especially in the Abstract where it appears multiple times.

We appreciate the reviewers' observation. We have carefully revised the manuscript to reduce the use of the word *putative*.

- Results:

o Paragraph 1 overview is a bit out of place here – that's more methods and detracts from the results

We thank the reviewers for this helpful suggestion. In line with their feedback, we have removed the textual overview of the study design from the beginning of the Results section and retained only the corresponding figure, which we believe provides sufficient context without detracting from the flow of the results. Following the suggestions of reviewer 1, we have added an overview of our methodology to the Introduction section.

o Expressions such as 'power bracket' are a bit odd; this sounds electrical almost, please amend to more conventional language used in the field.

We thank the reviewers for pointing this out. We have removed the term "power bracket" from our manuscript to improve clarity and alignment with standard terminology.

o Admixed Americans is quite a broad term and not a group that is represented in 1k genomes. Clarification is needed.

We thank the reviewers for their comment. We would like to clarify that Admixed American (AMR) is a standard term used by the 1000 Genomes Project to describe the AMR superpopulation, which includes individuals from populations such as Colombian, Mexican, Peruvian, and Puerto Rican ancestries (<https://mart.ensembl.org/info/genome/variation/species/populations.html>). To ensure clarity, we have revised the manuscript to explicitly refer to continental superpopulations, consistent with the 1000 Genomes terminology and definitions.

Addition to the Results section:

"Although genetic ancestry is on a continuous scale, here we use the continental superpopulations defined by the 1000 Genomes Project phase 3[46], i.e. individuals genetically similar to Africans (AFR), Europeans (EUR), East Asians (EAS), South Asians (SAS), and Admixed Americans (AMR)."

o In the section on different genetic ancestries lines 310 to 317 are discussion/interpretation, not results – please move this to the Discussion or take it out.

We thank the reviewers for this helpful suggestion. In line with their recommendation, we have moved the interpretation of the results to the Discussion section to maintain a clear distinction between both sections.

o Metformin results: we were a bit dubious about these results, as no numerical results/effect sizes are provided. The description of these results is a bit vague, and they appear to come out of nowhere with **little rationale provided**.

In line with the reviewers comments, and in agreement with reviewer 1's similar suggestion, we have removed the section discussing metformin from the revised manuscript.

o 193 – consistent language – etiology/causality are thrown around haphazardly.

Thank you for pointing this out. We have carefully reviewed the manuscript to revise or remove imprecise or inappropriate collocations throughout the text to improve clarity and precision in our language.

o 320 – figure 4 – we feel like this figure could do with being a 3x1 figure where the rows correspond to ancestries and the y-axis reflects the comorbidities.

We thank the reviewers for this helpful suggestion. We agree that restructuring the figure to emphasize the ancestry dimension of this section improves clarity. However, since the figure presents results for four ancestries and the meta-analysis, the revised layout is presented as a 5x1 figure.

Figure 4: Mendelian randomization (MR) results for different global genetic ancestry groups using cluster-stratified genetic predisposition for T2D and risk for T2D non-cardiovascular comorbidities (EUR=individuals genetically similar to Europeans, EAS=individuals genetically similar to East Asians) for the causal relationships (q -value < 0.05). Causal estimates are expressed as the odds ratio (OR) of each comorbidity per doubling (2-fold increase) in genetically determined dichotomous T2D risk. Filled circles mark robust estimates that passed all sensitivity analyses (T2DGGI = Type 2 Diabetes Global Genomics Initiative; CI = confidence interval; PI = proinsulin).

- Methods:

o Instrument selection criteria: what's the justification for the commonly employed p -value, r^2 and distance thresholds? There is no reference provided, and it is my understanding that there is no real consensus in the field as to how instruments should be selected.

We agree with the reviewers that there is no universally accepted standard for instrument selection criteria in MR studies. Hence, in the revised manuscript, we have removed the phrase "commonly employed" and have now added appropriate reference in the Methods section:

“Using all T2D index risk variants, our definition of independent IVs is not as strict as the one employed by the TwoSampleMR R package (v0.5.7), defined as LD-based clumped genome-wide significant (p -value $< 5 \times 10^{-8}$) variants with $r^2 < 0.001$ over a 10Mb window.”

[1] Hemani G, Zheng J, Elsworth B, Wade KH, Haberland V, Baird D, Laurin C, Burgess S, Bowden J, Langdon R, Tan VY, Yarmolinsky J, Shihab HA, Timpson NJ, Evans DM, Relton C, Martin RM, Davey Smith G, Gaunt TR, Haycock PC. The MR-Base platform supports systematic causal inference across the human phenome. *Elife*. 2018 May 30;7:e34408. doi: 10.7554/eLife.34408. PMID: 29846171.

o Can the authors provide a reference for the F-statistic method/calculation and >10 criterion?

Thank you for this important request. We have now added references to support our use of the F-statistic in the methods section. For the approximation of the F-statistic in cases when no individual level data is available, we have used the approximation formula derived in Bowden et al. (2018) which can be found in the appendix 2 [1].

[editorial note: redacted]

For the F-statistic threshold, we have referenced the Mendelian Randomization Primer by Sanderson et al. (2022) [2], which states that “As a general rule, if the first-stage F statistic is greater than 10, the level of this bias is small [3,4]. A cut-off of $F > 10$ has been used as a conventional threshold for a strong instrument in some studies.”

[1] Bowden, J., et al., Improving the accuracy of two-sample summary-data Mendelian randomization: moving beyond the NOME assumption. *International Journal of Epidemiology*, 2018. 48(3).

[2] Sanderson, E., et al., Mendelian randomization. *Nature Reviews Methods Primers*, 2022. 2(1).

[3] Staiger, D. & Stock, J. H. Instrumental variables regression with weak instruments. Report No. 0898-2937 (National Bureau of Economic Research, 1994).

[4] Stock, J. H. & Yogo, M. Testing for weak instruments in linear IV regression. Report No. 0898-2937 (National Bureau of Economic Research, 2002).

o Datasets: it's not clear what table is 'Table_T2DGGI_samplesize', is this supp table 9? If so, the ancestry groups are not the same in the table as in the text. The table has a Hispanic group, but no admixed group and the text is the other way round. Please clarify this and refer to the correct table or amend the table, if not. Admixed and Hispanic aren't the same.

We confirm that the table in question is Supplementary Table 9. We agree with the reviewers that “Hispanic” and “Admixed American” are not interchangeable. To address this inconsistency, we have updated the Supplementary Table 9 to refer to

the Admixed American group, as defined by the 1000 Genomes Project AMR continental super population.

o 266 – ‘evidence of putative causal links’ -> ‘indicating a multifactorial association’. ‘suggestions of supposed causal links’ which then ‘indicate’ such an association – the language throughout is too timid and hamstrings the impact of what is being said. We have revised the phrasing throughout the manuscript to convey our findings with greater clarity, while still reflecting the inherent limitations of causal inference using MR. Specifically, we have removed overly tentative expressions such as “putative”.

Updated sentence:

“We find evidence of a causal link between all T2DGGI genetic clusters, except the beta cell clusters, and CTS (Figure 3, SupTab1), demonstrating a multifactorial association mediated by diverse biological pathways.”

o Using Plink 1.9 also odd, do the results differ at all using the updated software, ie Plink 2.0?

We thank the reviewers for raising this point. The clumping methodology implemented in PLINK 1.9 and PLINK 2.0 is functionally equivalent, as both use the same core algorithm for LD-based clumping. We nonetheless conducted a comparison between the two versions to ensure consistency and observed no differences in the set of clumped variants across T2D comorbidities. These results confirm that our findings are robust to the version of PLINK used. We would like to highlight that PLINK 2.0 is much faster than PLINK 1.9. We have updated the Methods section accordingly.

o Any consideration given to analysing vascular and all-cause dementia?

We appreciate the reviewers’ suggestion. We considered including both vascular and all-cause dementia in our analysis. However, we ultimately decided against it due to our previous selection criteria for including T2D comorbidities GWAS, which we emphasize in the revised Results section:

“The T2D comorbidities were selected after a comprehensive literature review of observational studies and the availability of GWAS data that did not include proxy cases.”

Specifically, for all-cause dementia, the largest available GWAS [1,2] include proxy cases based on family history (i.e., "GWAX" designs), as well as non–age-matched controls. This approach has been shown to introduce bias and reduce the accuracy of effect estimates. Notably, studies have demonstrated significant differences in genetic associations and inferred causal relationships when comparing GWAX to GWAS based on clinically diagnosed cases [3]. Given these concerns, and to maintain methodological consistency with our other trait selections, we opted to exclude all-cause dementia GWAS that rely on proxy data.

We have added these considerations to the Methods section of our updated manuscript:

“Despite being observationally associated with T2D, we have not included Parkinson’s disease nor all-cause dementia to our study due to potential bias from proxy-case GWAS.”

Regarding vascular dementia, the largest GWAS summary statistics to date are not publicly available [4]. However, we have reached out to the authors, who gave us access to the full GWAS summary statistics. The revised manuscript, figures and supplemental material now include the results for vascular dementia as a further T2D comorbidity. Briefly, we identify a risk-increasing effect of overall T2D genetic predisposition on vascular dementia (OR=1.09, q-value=1.12x10⁻²) driven by the obesity cluster (OR=1.18, q-value=9.96x10⁻³) and beta cell cluster associated with increased proinsulin levels (OR=1.13, q-value=3.20x10⁻²). Similarly to the Alzheimer’s disease results, we identify a protective effect of genetic predisposition for vascular dementia on T2D risk (OR=0.95, q-value=1.35x10⁻³⁶), which gets attenuated upon adjustment for BMI, WHR or HDL levels.

[1] Bellenguez C, Küçükali F, Jansen IE, et al. New insights into the genetic etiology of Alzheimer's disease and related dementias. *Nat Genet.* 2022 Apr;54(4):412-436. doi: 10.1038/s41588-022-01024-z. PMID: 35379992.

[2] Wightman DP, Jansen IE, Savage JE, et al. A genome-wide association study with 1,126,563 individuals identifies new risk loci for Alzheimer's disease. *Nat Genet.* 2021 Sep;53(9):1276-1282. doi: 10.1038/s41588-021-00921-z. PMID: 34493870.

[3] Wu Y, Sun Z, Zheng Q, et al. Pervasive biases in proxy genome-wide association studies based on parental history of Alzheimer's disease. *Nat Genet.* 2024 Dec;56(12):2696-2703. doi: 10.1038/s41588-024-01963-9. PMID: 39496879.

[4] Mega Vascular Cognitive Impairment and Dementia (MEGAVCID) consortium. A genome-wide association meta-analysis of all-cause and vascular dementia. *Alzheimers Dement.* 2024 Sep;20(9):5973-5995. doi: 10.1002/alz.14115. PMID: 39046104.

- Discussion:

o Metformin findings: we think that the way this is written is a bit of a reach re saying that the authors have identified it as a potential repurposing opportunity for osteoarthritis. This needs softening and more details of the actual results, as highlighted above. This also needs to be triangulated and replicated with other approaches, as MR is not the only method used to ascertain causality.

We have removed the metformin analysis from the updated version of our manuscript following the reviewer’s suggestions.

o 412: the authors declare that they were limited by power, but at no point have they conducted formal power analyses (there are calculators for this for MR studies) and looking at the number of cases in the T2DGGI GWAS they look reasonable...perhaps

they can tone this down a little, as they don't know for sure that they were underpowered?

We have revised the manuscript to tone down statements related to limited statistical power.

Revised Discussion section:

“The sample size of the T2DGGI GWAS meta-analysis used to identify T2DGGI genetic risk variants and create the mechanistic clusters is much larger than some of the non-cardiovascular comorbidity GWAS used in this work. This may influence the power to detect causal effects in the reverse MR analysis.”

o We think a bit more discussion is needed around certain results, e.g. Alzheimer's protective for diabetes...

We thank the reviewers for their suggestion. To further dissect the identified protective effect of Alzheimer's disease and vascular dementia on T2D risk, we have conducted additional multivariable MR analysis to adjust the causal effect estimates for BMI, WHR and HDL levels.

Addition to the Results section:

“The identified protective effects of Alzheimer's disease and vascular dementia genetic predisposition on T2D risk get attenuated after adjustment through multivariable MR for high-density lipoprotein (HDL) levels, body mass index (BMI) and waist-to-hip ratio (WHR) (Figure 1, Methods, Supplemental Note, SupTable6).”

We have added our interpretation of the results to the revised Discussion section:

“We identify protective effects of genetic predisposition for Alzheimer's disease and vascular dementia on T2D risk that do not show evidence of horizontal pleiotropy. For vascular dementia, the reverse MR analysis was driven primarily by a single genome-wide significant variant at the APOE locus, a well-established vertical pleiotropy region with roles in both lipid metabolism and neurodegeneration[76]. The protective effect observed may thus reflect APOE-mediated lipid regulation. This is further supported by the attenuation of the identified effect after adjustment for genetic predisposition for higher HDL levels, which suggests that the effect of genetic predisposition for both dementia types on T2D risk is at least partially explained by HDL levels. (...) Our findings highlight that lipid metabolism pathways may influence risk of dementia and T2D in different directions.”

We have also extended our discussion regarding the divergent T2DGGI cluster-stratified effects identified to be associated with osteoarthritis risk:

“Previous MR studies have not found evidence of a causal link between T2D genetic predisposition and osteoarthritis[54, 55]. By leveraging the largest and most recent GWAS datasets for both diseases, we are now able to identify a significant risk-increasing effect of T2D genetic predisposition on osteoarthritis. This effect is attenuated after adjusting for adiposity-related traits such as BMI, WHR, and subcutaneous adipose tissue volume. This indicates that these shared risk factors

explain at least partially the effect of T2D genetic predisposition on osteoarthritis risk. We also observe contrasting cluster-specific effects: genetic variants from the T2DGGI obesity cluster have a risk-increasing effect on osteoarthritis, while variants assigned to beta cell function-related T2DGGI clusters show a protective effect on osteoarthritis risk. These findings align with prior observations of opposing associations at shared genetic loci between the two conditions[55], and highlight the value of mechanistic decomposition of polygenic risk.”